# Mutual Information Divergence: A Unified Metric for Multimodal Generative Models

**Jin-Hwa Kim**[*]
NAVER AI Lab, SNU AIIS
Republic of Korea
j1nhwa.kim@navercorp.com

**Yunji Kim**    **Jiyoung Lee**
NAVER AI Lab
Republic of Korea
{yunji.kim,lee.j}@navercorp.com

**Kang Min Yoo**
NAVER AI Lab, CLOVA, SNU AIIS
Republic of Korea
kangmin.yoo@navercorp.com

**Sang-Woo Lee**
NAVER CLOVA, AI Lab, KAIST AI
Republic of Korea
sang.woo.lee@navercorp.com

## Abstract

Text-to-image generation and image captioning are recently emerged as a new experimental paradigm to assess machine intelligence. They predict continuous quantity accompanied by their sampling techniques in the generation, making evaluation complicated and intractable to get marginal distributions. Based on a recent trend that multimodal generative evaluations exploit a vison-and-language pre-trained model, we propose the negative Gaussian cross-mutual information using the CLIP features as a unified metric, coined by Mutual Information Divergence (MID). To validate, we extensively compare it with competing metrics using carefully-generated or human-annotated judgments in text-to-image generation and image captioning tasks. The proposed MID significantly outperforms the competitive methods by having consistency across benchmarks, sample parsimony, and robustness toward the exploited CLIP model. We look forward to seeing the underrepresented implications of the Gaussian cross-mutual information in multimodal representation learning and future works based on this novel proposition. The code is available at https://github.com/naver-ai/mid.metric.

## 1   Introduction

A multimodal generative model, including text-to-image generation [1] and image captioning [2] models, is an emerging research topic showing interpretative multimodal understanding, text or image retrieval, machine creativity, *etc*. The gist of learning multimodal generative models is to understand how to connect one modality to the other and generate the corresponding representations following the desired data distribution. However, measuring the distance or divergence between the model and data distributions is generally intractable due to the finite data and generation cost. Therefore, the proposed metrics attempt to approximate it with polynomial-sized data and generated samples [3].

For text-to-image generation, the widely-used metrics are Inception Score (IS) [4] and Fréchet Inception Distance (FID) [5]. These metrics are originally proposed for non-conditional generative models, which are repurposed to measure the distance between the data and model conditional distributions using a validation split. This idea was supported by the effectiveness of deep features as a perceptual metric [6], although these metrics use the Inception V3 [7]. Not surprisingly, there are attempts to develop more robust metrics using multimodal pre-trained models, such as object

---

[*]Corresponding author.

36th Conference on Neural Information Processing Systems (NeurIPS 2022).

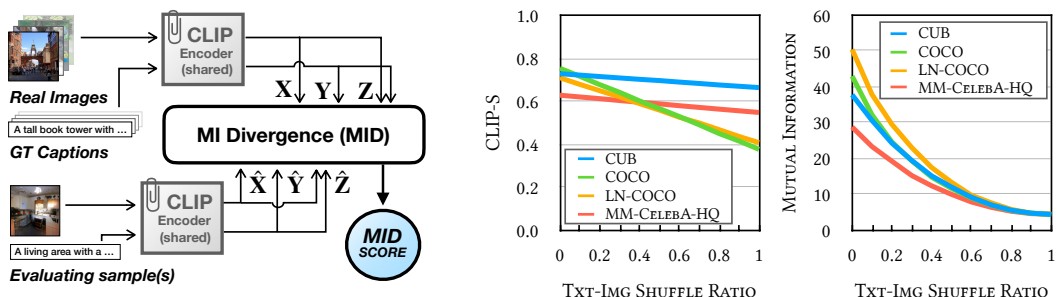

Figure 1: **Left.** The schematic diagram of our proposed method. **Right.** The consistent property of Gaussian MI is in stark contrast to CLIP-S [19]. We vary the ratio of text-image shuffling where the counterparts in the selected pairs are deliberately shuffled, depicting misalignment. For CLIP-S, we observe varying slopes across the datasets depending on the data domain while MI shows a relatively consistent tendency. The CUB and MM-CelebA-HQ describe birds and human faces, respectively, having narrow domains compared to COCO and LN-COCO with various objects.

detectors [8], image captioning models [9], and vision-and-language pre-trained models [10, 11]. Significantly, the image captioning method transforms text-to-image measurement into image-to-text measurement, providing a different viewpoint with cyclic consistency.

For image-to-text generation, or image captioning, the COCO Caption Evaluation toolkit seems to be the standard to measure the divergence from ground-truth captions having BLEU [12], METEOR [13], ROUGE [14], CIDEr [15], and SPICE [16]. Similar to the IS and FID of text-to-image generation metrics, these metrics are merely n-gram-based statistical methods neglecting conditional images. The breakthrough in improving the correlation with human judgment comes from the utilization of the pre-trained vision-and-language models, e.g., TIGEr [17], ViLBERTScore-F [18], and RefCLIP-S [19]. We speculate that these two directional metrics are getting closer to measuring the generative divergence of text-image alignment.

This paper proposes a unified metric for multimodal generative models. In probability theory and information theory, mutual information (MI) measures how much one random variable tells us about the other. In a multimodal generation, the MI of two modalities quantitatively measures how much the generated is well-aligned with the condition. From this motive, we propose to use the Gaussian mutual information where the probability distributions are defined by the means and covariances of visual and textual features and borrow the idea of cross-mutual information [20] to measure the MI divergence from the real data distribution, which is the expectation of point-wise mutual information with respect to evaluating samples. Surprisingly, the proposed method outperforms previous works with significant margins on the assorted benchmarks of text-to-image generation and image captioning, including standard human judgment correlation benchmarks.

Section 2 introduces the previous works on text-to-image generation and image captioning metrics, and the work on cross-mutual information. Section 3 describes the proposed method defining the continuous mutual information using multivariate Gaussian distributions and the negative cross-mutual information, which is the Mutual Information Divergence (MID) that we term. Section 4 consists of two parts, evaluation on text-to-image generation and image captioning evaluation, including related discussions. Section 6 concludes the work with remarks.

We summarize our contributions as follows:

- To the best of our knowledge, we firstly propose the negative cross-mutual information under the Gaussian assumption as a unified metric for multimodal generative models.
- We provide three theoretical analyses on the proposed method MID, out-of-distribution detecting by the squared Mahalanobis distances, bias and variance decomposition, and its relation to the Kullback-Leibler divergence.
- We achieve the state-of-the-art in text-to-image generation and image captioning benchmarks including the generated and human Likert-scale judgment correlations, visual reasoning accuracy, Flickr8K-Expert, Flickr8K-CF, Pascal-50S, and FOIL hallucination detection.

## 2  Related work

### 2.1  Metrics for assessing text-to-image generation

**Traditional metrics.** One of the widely used metrics is the Fréchet Inception Distance (FID) [5] that measures the distributional difference between synthetic (fake) and real-world images (real). Training and validation distributions are independent, making a model that fails to match the conditional distributions if it does not reflect the given textual information. Although it measures fidelity along with Inception Score [4], it cannot directly measure the alignment of text and image. The alternative metrics [21–23] are proposed to evaluate the fidelity and diversity. **Text-to-image metrics.** Dedicated to assessing text-to-image generation, the R-Precision exploits the Deep Attentional Multimodal Similarity Model (DAMSM) [24] to calculate the top-1 retrieval accuracy from one hundred text candidates for the generated image as a query. Besides, the CLIP R-Precision [11] exploits the CLIP [10] showing a better retrieval performance and human judgment correlation. However, false-negative candidates (accidentally correlated) or strong negative candidates (totally unrelated) may interfere with the accurate assessment [25]. To evaluate the quality of individual objects, the SOA [8] attempts to measure the object detection accuracy using the YOLOv3 [26] based on the object classes that appeared in the text but cannot consider other factors. Caption generation [9] is another approach using the vision-and-language pre-trained model. The motivation is the cyclic consistency that the generated caption from the generated image should match with the text for image generation. However, the model bias including object hallucination [27] and the accumulated errors from metrics are drawbacks. **Diagnostic datasets.** Park et al. [11] provide the curated splits of the CUB [28, 29] and Flowers [30] to assess unseen color and shape compositions in the narrow domains. DALL-Eval [31] proposed a diagnostic dataset *PaintSkills* to evaluate visual reasoning skills to assess models based on this dataset. Since the dataset is generated from a 3D simulator using limited configurations, the data distributions deviate from other real-world datasets.

### 2.2  Metrics for assessing image captioning

**Reference-only metrics.** Borrowing from machine translation literature, image captioning models are evaluated using reference-based textual metrics where five references are usually used. BLEU-4 [12], ROUGE-L [14], and METEOR [13] are n-gram precision or recall-based metrics, CIDEr [15] uses tf-idf weighting and stemming, while SPICE [16] uses semantic parsing and scene graph analysis. Notably, BERT-S++ [32] considers inter-reference variance using the fine-tuned BERTScore [33] for image captioning. **Reference-with-image metrics.** The recently proposed metrics are considering the images used for generating captions. TIGEr [17] uses a pre-trained SCAN [34] while ViLBERTScore-F [18] uses a pre-trained ViLBERT [35] exploiting the vision-and-language alignments from large-scale data and multiple tasks. Similarly, CLIP-S and RefCLIP-S [19] use the pre-trained CLIP, a more powerful vision-and-language model, outperforming the previous methods. **Implications.** CLIP R-Precision for text-to-image generation and RefCLIP-S for image captioning share the same motivation exploiting the same vision-and-language pre-trained model. Here, we remark on the unifying metrics in two different tasks (*e.g.*, CLIP-S), text-to-image generation and image captioning, and propose a new unified metric for the multimodal generative models based on the continuous mutual information considering the covariances of two modality groups.

### 2.3  Metrics for assessing other tasks

One of the other applications worth mentioning is text-based motion generation tasks [36]. Since traditional metrics, *e.g.*, average position error or average variance error, merely rely on numerical differences instead of high-level semantics, they proposed to use a motion-text alignment pre-trained model (mCLIP), enabling us to measure CLIP-S and our proposed MID. It validates our speculation that the unified metric can effectively apply beyond vision and language.

### 2.4  Cross-mutual information in machine translation

Bugliarello et al. [20] proposed the cross-mutual information (XMI) as a metric of machine translation exploiting a probabilistic view in neural machine translation models. XMI is an analogue of mutual information for cross-entropy defined as $\text{XMI}(S \rightarrow T) = H_{q_{\text{LM}}}(T) - H_{q_{\text{MT}}}(T|S)$. $H_{q_{\text{LM}}}(T)$ denotes the cross-entropy of the target sentence $T$ under a language model $q_{\text{LM}}$ and $H_{q_{\text{MT}}}(T|S)$ is the cross-

conditional entropy under a cross-lingual model $q_{\text{MT}}$. In practice, they exploit two model distributions $q_{\text{LM}}(\mathbf{t})$ and $q_{\text{MT}}(\mathbf{t}|\mathbf{s})$ to approximate the XMI as follows:

$$\text{XMI}(S \rightarrow T) \approx -\frac{1}{N} \sum_{i=1}^{N} \log \frac{q_{\text{LM}}(\mathbf{t}^{(i)})}{q_{\text{MT}}(\mathbf{t}^{(i)}|\mathbf{s}^{(i)})}$$

where $N$ denotes the number of held-out evaluating samples. Since language models consider a finite size of vocabulary, the cross-entropy can be efficiently approximated using the target sentences; however, it is limited to readily applying to other generative models.

## 3 Method

### 3.1 Continuous mutual information

We introduce a unified metric for conditional generative models not depending on the modalities of *condition* and *generation*. To measure the alignment of condition and generation, we first consider the continuous mutual information of the condition $\mathbf{x}$ and generation $\mathbf{y}$ as follows:

$$I(\mathbf{X}; \mathbf{Y}) = \mathbb{E}_{p(\mathbf{x},\mathbf{y})} \log \frac{p(\mathbf{x},\mathbf{y})}{p(\mathbf{x})p(\mathbf{y})} \tag{1}$$

where the probability and joint probability distributions are multivariate Gaussian, which is the maximum entropy distribution for the given mean $\mu$ and covariance $\Sigma$ [37]. The first two moments are used for practical reason [5]. The multivariate Gaussian distribution is defined as:

$$p(\mathbf{x}) = \frac{1}{\sqrt{(2\pi)^D \det(\Sigma)}} \exp\left[-\frac{1}{2}(\mathbf{x} - \mu)^{\mathsf{T}} \Sigma^{-1} (\mathbf{x} - \mu)\right] \tag{2}$$

where $D$ is the dimension of $\mathbf{x}$. The mutual information with the Gaussian distributions is reduced to:

$$I(\mathbf{X}; \mathbf{Y}) = \frac{1}{2} \log \left( \frac{\det(\Sigma_{\mathbf{x}}) \det(\Sigma_{\mathbf{y}})}{\det(\Sigma_{\mathbf{z}})} \right) \tag{3}$$

where $\Sigma_{\mathbf{x}}$ and $\Sigma_{\mathbf{y}} \in \mathbb{R}^{D \times D}$ are the covariances of the condition and generation and $\Sigma_{\mathbf{z}} \in \mathbb{R}^{2D \times 2D}$ is the covariance matrix of the concatenation $\mathbf{z}$ of $\mathbf{x}$ and $\mathbf{y}$ representing the joint distribution. The proof can be found in Appendix A.1. Note that we use $\log \det(\Sigma) = \sum_i \log \lambda_i$ for numerical stability, where $\lambda_i$ is the eigenvalue of $\Sigma$.

For high-dimensional features, we consider two encoders $f_{\mathbf{x}}$ and $f_{\mathbf{y}}$ to get $\tilde{\mathbf{x}}$ and $\tilde{\mathbf{y}}$, respectively, maximizing the mutual information of the encoded representations, $I(\tilde{\mathbf{X}}; \tilde{\mathbf{Y}})$. Specifically, the image and text encoders of the CLIP [10] are used since these are pre-trained on 400M image-text pairs using the InfoNCE loss, maximizing a lower bound on mutual information [38]. Without loss of generality, we use $\mathbf{x}, \mathbf{y}$ instead of $\tilde{\mathbf{x}}, \tilde{\mathbf{y}}$ to denote the feature vectors to calculate the moments.

### 3.2 Point-wise mutual information for pair-wise evaluation

Based on the previous continuous mutual information, we derive the *point-wise mutual information (PMI)* for pair-wise evaluation. Please see Appendix A.1 for the detail. The PMI is defined as:

$$\text{PMI}(\mathbf{x}; \mathbf{y}) = I(\mathbf{X}; \mathbf{Y}) + \frac{1}{2} \left( D_M^2(\mathbf{x}) + D_M^2(\mathbf{y}) - D_M^2(\mathbf{z}) \right). \tag{4}$$

where $D_M^2$ denotes the squared Mahalanobis distance (SMD), $D_M^2(\mathbf{x}) = (\mathbf{x} - \mu_{\mathbf{x}})^{\mathsf{T}} \Sigma_{\mathbf{x}}^{-1} (\mathbf{x} - \mu_{\mathbf{x}})$ where $\mu_{\mathbf{x}}$ and $\Sigma_{\mathbf{x}}$ are the mean and covariance of $\mathbf{x}$, and similarly for $D_M^2(\mathbf{y})$ and $D_M^2(\mathbf{z})$. $\mathbf{z}$ denotes $[\mathbf{x}; \mathbf{y}]$. The MI is from the normalization of the Gaussians and the SMDs are from the exponential of the Gaussians. The second term measures the difference between the distances, $D_M^2(\mathbf{x}) + D_M^2(\mathbf{y})$ and $D_M^2(\mathbf{z})$, assessing the deviation from the MI. Notice that the expectation of the second term with respect to the sample distribution is zero (see Appendix A.1).

Table 1: Generated Likert-scale judgment correlation using LAFITE. † uses the i.i.d. samples having at least one detected object, which was 88.3% of samples, to calculate the SOA accuracy per image.

| Method | Backbone | Kendall $\tau_c$ | Kendall $\tau_b$ |
|---|---|---|---|
| SOA$^\dagger$ [8] | YOLO-V3 | 47.3 | 51.8 |
| CLIP-S [19] | CLIP (ViT-B/32) | 40.8 | 35.3 |
| InfoNCE [38] | CLIP (ViT-B/32) | 44.1 | 38.2 |
| CLIP-R-Precision [11] | CLIP (ViT-B/32) | 66.0 | 56.1 |
| OFA-Captioning+CLIP-S [19, 39] | OFA-Large + CLIP (ViT-B/32) | 72.0 | 62.3 |
| CLIP-S [19] | CLIP (ViT-L/14) | 52.2 | 45.2 |
| InfoNCE [38] | CLIP (ViT-L/14) | 64.8 | 56.1 |
| CLIP-R-Precision [11] | CLIP (ViT-L/14) | 69.6 | 58.1 |
| OFA-Captioning+CLIP-S [19, 39] | OFA-Large + CLIP (ViT-L/14) | 73.7 | 63.8 |
| MID (ours) | CLIP (ViT-B/32) | 74.6 | 64.6 |
| MID (ours) | CLIP (ViT-L/14) | **87.3** | **75.6** |

### 3.3 Mutual Information Divergence: the expectation of PMI w.r.t evaluating samples

We propose to use the expectation of PMI with respect to the evaluating sample $(\hat{\mathbf{x}}, \hat{\mathbf{y}})$, measuring the divergence from the ground-truth or reference samples $(\mathbf{X}, \mathbf{Y})$. The metric is defined as follows:

$$\mathbb{E}_{(\hat{\mathbf{x}},\hat{\mathbf{y}})\sim\mathcal{D}}\text{PMI}(\hat{\mathbf{x}};\hat{\mathbf{y}}) = I(\mathbf{X};\mathbf{Y}) + \frac{1}{2}\mathbb{E}_{(\hat{\mathbf{x}},\hat{\mathbf{y}})\sim\mathcal{D}}\big[D_M^2(\hat{\mathbf{x}}) + D_M^2(\hat{\mathbf{y}}) - D_M^2(\hat{\mathbf{z}})\big]. \tag{5}$$

where $\mathcal{D}$ and $(\hat{\mathbf{x}}, \hat{\mathbf{y}})$ denote the set of evaluating samples and a pair of evaluating sample, respectively, $\hat{\mathbf{z}}$ denotes $[\hat{\mathbf{x}};\hat{\mathbf{y}}]$. Notice that the expectation of $D_M^2(\hat{\mathbf{x}})$ can be decomposed to the bias and variance terms as follows (Appendix A.2 for the proof):

$$\mathbb{E}_{\hat{\mathbf{x}}}\big[D_M^2(\hat{\mathbf{x}})\big] = (\mu_{\hat{\mathbf{x}}} - \mu_{\mathbf{x}})^\intercal \Sigma_{\mathbf{x}}^{-1}(\mu_{\hat{\mathbf{x}}} - \mu_{\mathbf{x}}) + \text{tr}\big(\Sigma_{\mathbf{x}}^{-1}(\Sigma_{\hat{\mathbf{x}}} - \Sigma_{\mathbf{x}})\big) + D \tag{6}$$

considering the mean and covariance deviations from the reference, $\mathcal{N}(\mu_{\mathbf{x}}, \Sigma_{\mathbf{x}})$. $\mathbb{E}_{\hat{\mathbf{y}}}\big[D_M^2(\hat{\mathbf{y}})\big]$ is $D$ when $\hat{\mathbf{y}} = \mathbf{y}$ as a generative condition since the two moments are equal to the counterparts.

By the way, we can show that $\mathbb{E}_{(\hat{\mathbf{x}},\hat{\mathbf{y}})\sim\mathcal{D}}\text{PMI}(\hat{\mathbf{x}};\hat{\mathbf{y}})$ is related to the Kullback–Leibler divergence as follows (The proof can be found in Appendix A.3):

$$\mathbb{E}_{(\hat{\mathbf{x}},\hat{\mathbf{y}})\sim\mathcal{D}}\text{PMI}(\hat{\mathbf{x}};\hat{\mathbf{y}}) = I(\hat{\mathbf{X}};\hat{\mathbf{Y}}) + D_{\text{KL}}(p(\hat{\mathbf{x}}) \parallel p(\mathbf{x})) - D_{\text{KL}}(p(\hat{\mathbf{z}}) \parallel p(\mathbf{z})) \tag{7}$$

For simplicity, we denote our proposed method $\mathbb{E}_{(\hat{\mathbf{x}},\hat{\mathbf{y}})\sim\mathcal{D}}\text{PMI}(\hat{\mathbf{x}};\hat{\mathbf{y}})$ as **Mutual Information Divergence (MID)**, comparable of the FID. In practice, we use Equation 5 using the double-precision CLIP features. For point-wise evaluation, we use the $\text{PMI}(\hat{\mathbf{x}}, \hat{\mathbf{y}})$ without the expectation.

## 4 Experiment

### 4.1 Evaluation on text-to-image generation

**Implementation details.** Without an explicit mention, we use the CLIP (ViT-L/14) to extract image and text embedding vectors. Note that it is crucial to use double-precision for numerical stability.

**Generated Likert-scale judgments.** To carefully assess the text-image alignment, we consider the four-scale alignment using the real and fake images from the COCO dataset [2]. We regard the real images as a four-point set, the fake images generated by the ground-truth captions as a three-point set, the fake images generated by the foiled captions [2] [40] as a two-point set, and the randomly sampled (misaligned) fake images as one-point set. We assume that the fake images generated by the foiled captions should be inferior compared with the fake images generated by the ground-truth captions because the model cannot exploit the critical information to generate key objects. The current state-of-the-art LAFITE [41] pre-trained on the COCO [3] is used for our text-to-image generation

---

[2]For the details, please refer to the object hallucination section in Section 4.2 and Figure 11 in Appendix.
[3]https://github.com/drboog/Lafite

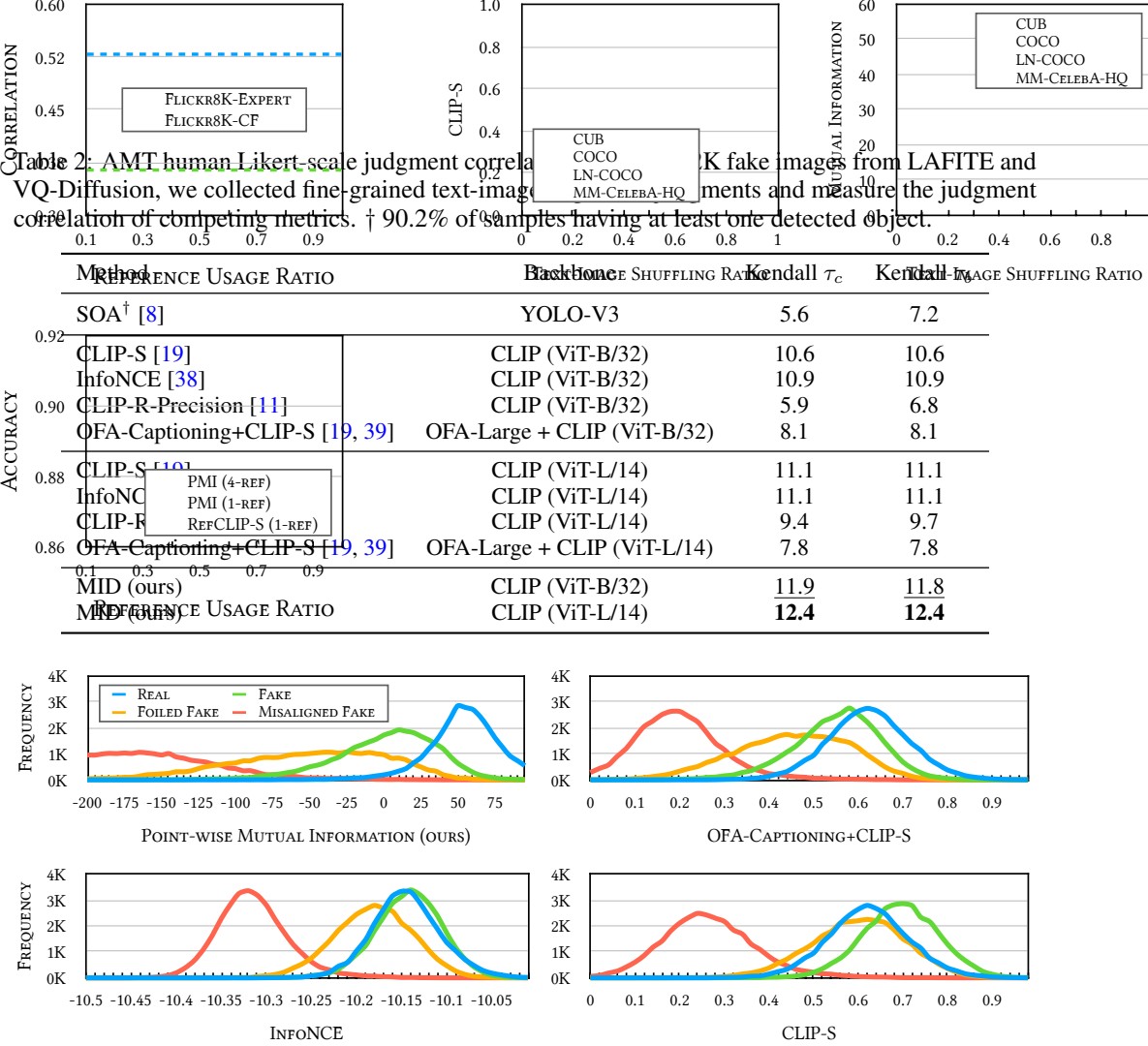

Table 2: AMT human Likert-scale judgment correla... K fake images from LAFITE and VQ-Diffusion, we collected fine-grained text-imag... ments and measure the judgment correlation of competing metrics. † 90.2% of samples having at least one detected object.

| Method | Backbone | Kendall $\tau_c$ | Kendall-in |
|---|---|---|---|
| SOA† [8] | YOLO-V3 | 5.6 | 7.2 |
| CLIP-S [19] | CLIP (ViT-B/32) | 10.6 | 10.6 |
| InfoNCE [38] | CLIP (ViT-B/32) | 10.9 | 10.9 |
| CLIP-R-Precision [11] | CLIP (ViT-B/32) | 5.9 | 6.8 |
| OFA-Captioning+CLIP-S [19, 39] | OFA-Large + CLIP (ViT-B/32) | 8.1 | 8.1 |
| CLIP-S [19] | CLIP (ViT-L/14) | 11.1 | 11.1 |
| InfoNCE [38] | CLIP (ViT-L/14) | 11.1 | 11.1 |
| CLIP-R-Precision [11] | CLIP (ViT-L/14) | 9.4 | 9.7 |
| OFA-Captioning+CLIP-S [19, 39] | OFA-Large + CLIP (ViT-L/14) | 7.8 | 7.8 |
| MID (ours) | CLIP (ViT-B/32) | _11.9_ | _11.8_ |
| MID (ours) | CLIP (ViT-L/14) | **12.4** | **12.4** |

Figure 2: The histogram shows the frequencies of the competing methods using the CLIP of ViT-L/14 for the four-scale judgments from the COCO dataset. For the detail of method, please see the text.

model. Notice that when we evaluate the metrics, the ground-truth captions are used to measure the text-image alignment of the generated images. We believe this generated benchmark can be a proxy to human judgments with a careful manipulation of the fake images using the foiled captions.

We randomly sample 30K captions from the FOIL dataset [40] to build 120K judgments. We report the Kendall's $\tau$ coefficient [42] to measure the rank correlation, a variant of $\tau_c$ or $\tau_b$ accounting for ties. In Table 1, our method consistently outperforms competing methods. InfoNCE denotes the negative InfoNCE loss [38] calculating the softmax function over the 30K captions. Remind that the InfoNCE maximizes a lower bound on mutual information [38]. Since this is estimated using a smaller batch size when optimizing, it shows a limited capability as a metric for the text-image alignment. Please refer to Table 8 and Figure 8 in Appendix for the VQ-Diffusion [43] benchmark.

Figure 2 (top left) shows the histogram of the frequencies of PMI showing the four categories. **Fake** samples have generally lower values than **Real**'s. **Foiled Fake** has lower values than **Real** and **Fake** having a long tail. We observe that foiled caption broadly impacts the text-image alignment as we expected. The negative PMI is often observed for the fake images, which are deviated from the distribution of real images. Whereas, the histogram of CLIP-S [4] (bottom right) and InfoNCE (bottom left) show **Fake** samples where its scores are higher than **Real**'s, while the overlapping areas of **Fake** and **Foiled Fake** (62.2% and 61.6% for CLIP-S and InfoNCE, respectively) are greater than PMI's (55.2%), making it difficult to differentiate the degree of text-image alignment. For the caption generation method [9], we exploit the current state-of-the-art image captioning model of the OFA-Large [39] pre-trained on a huge mixture of publicly available datasets to generate captions for

---

[4]The CLIP-S [19] is defined as $2.5 \times \cos(\mathbf{x}, \mathbf{y})$ which is the scaled cosine similarity of the CLIP features.

Table 3: Visual reasoning accuracy using the foiled caption trick.

| Metric | Object | Count | Color | Spatial | Object | Count | Color | Spatial |
| --- | --- | --- | --- | --- | --- | --- | --- | --- |
| | | CLIP ViT-B/32 | | | | CLIP ViT-L/14 | | |
| CLIP-S [19] | 0.318 | 0.026 | 0.068 | 0.025 | 0.585 | 0.157 | 0.209 | 0.169 |
| CLIP-R-Precision [11] | 0.168 | 0.016 | 0.031 | 0.019 | 0.238 | 0.046 | 0.058 | 0.041 |
| SOA [8] | 0.367 | 0.028 | 0.039 | 0.035 | 0.365 | 0.030 | 0.035 | 0.030 |
| InfoNCE [38] | 0.416 | 0.042 | 0.094 | 0.049 | 0.675 | 0.230 | 0.284 | 0.243 |
| MID (ours) | **0.792** | **0.290** | **0.332** | **0.280** | **0.843** | **0.443** | **0.481** | **0.457** |

the generated images. Then, CLIP-S [19] is used to assess the quality of image captioning. Notice that CLIP-S outperforms traditional image captioning metrics as shown in Section 4.2.

**Visual reasoning accuracy using the foiled caption trick.** Inspired by the foiled caption trick, we extend to four visual reasoning skills for object, count, color, and spatial relationship. For each category, we define a set of tokens, and we foiled those tokens in the caption by randomly swapping to the other token. We build three sets of images, the real images, the fake images, and the foiled fake images. We measure the accuracy that getting one point for the foiled fake images having the lowest score, or zero for the other cases. Table 3 shows that our method achieved the best performance across all categories. The runner-up was InfoNCE, while SOA was ineffective to differentiate among the two fake images and the real image. Although DALL-Eval [31] proposed to use a detector and its dedicated heads for the count, color, and spatial relationship tasks, this was limited to the 3D-generated images with their near-perfect detection ability. Remind that the accuracy of random guessing is 33.3%, where MID requires a powerful feature extractor of the CLIP ViT-L/14 to get meaningful performances on the count, color, and spatial relationship tasks. For the detail, please refer to the text in Appendix C.

**Human Likert-scale judgment.** We collect 10K one-to-four Likert-scale human judgments for 2K fake images using the LAFITE and VQ-Diffusion from the Amazon Mechanical Turk (AMT) for the fine-grained comparison with the competing metrics. For each image, we collect five annotations from unique workers taking its median for a reliable correlation measurement. Table 2 shows the consistent results with our generated judgment correlation benchmark. Since this benchmark aims for the fine-grained judgment among only fake images, overall scores are relatively lower than the generated benchmark. All correlation results are significant having the p-value < 0.001. For the details of the collection procedure and data statistics, please refer to Appendix D, where Figure 10 shows the visualization of some examples comparing with human judgment scores. We also report the comp-t2i benchmark [11] results for the compositional evaluation of the CUB and Flower datasets in aspects of color and shape in Appendix F.

### 4.1.1 Discussions

**Consistent metric across datasets.** A distinct property of our method is the consistency across datasets. As shown in Figure 1, the cosine similarity-based method, CLIP-S suffers the inconsistent results. For example, the CUB [28, 29] and MM-CelebA-HQ [44] have narrow domains, birds and human faces, respectively, which is prone to get a similar score for all samples in the datasets by cosine similarity. To validate our hypothesis, we vary the ratio of text-image shuffling using the real datasets where the counterparts in the selected pairs are deliberately shuffled, depicting misalignment. For CLIP-S, we observe the inconsistency depending on datasets, while MI shows a consistent tendency. Notice that the expectation of PMI is reduced to MI for the real images (Appendix A.1).

**Inspecting possible over-fitting with the CLIP features.** Figure 3 shows the normalized scores of metrics across feature extractors. To get the normalized score, we subtract the score of RN101 and divide by the standard deviation of four scores across feature extractors reminiscent of z-score. We expect the *footprint* of metrics should be consistent across different generative models if the model is not over-fitted to the metrics. LAFITE used both encoders of CLIP ViT-B/32 and VQ-Diffusion used the text encoder of CLIP ViT-B/32, while DM-GAN did none of them [45]. InfoNCE and CLIP-R-Precision are related to the contrastive training losses (LAFITE, and DM-GAN for the DAMSM loss [24]), which may lead to drastic change of the normalized scoring signature across the feature extractors. While the proposed MID was relatively stable across the generative models.

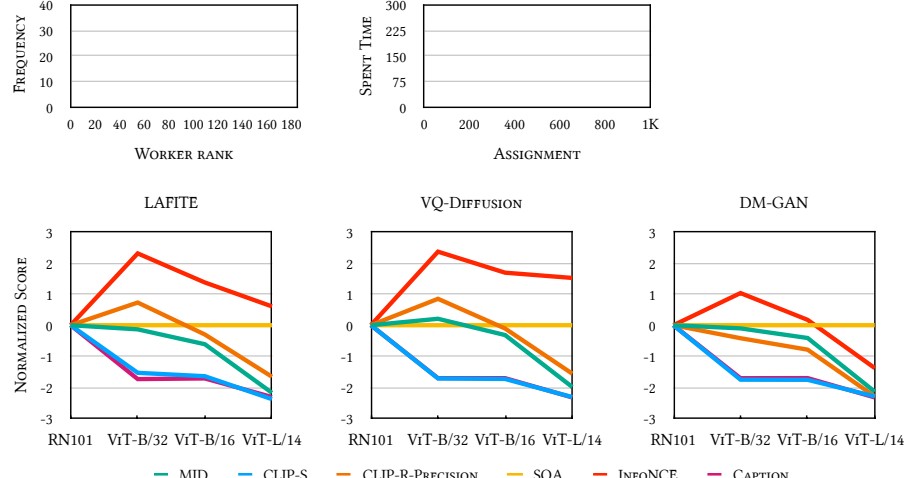

Figure 3: The footprints of metrics across feature extractors using the normalized scores.

Table 4: Flickr8K-Expert human judgment correlation.

| Method | Kendall $\tau_c$ |
|---|---|
| BLEU-1 [12] | 32.3 |
| BLEU-4 [12] | 30.8 |
| ROUGE-L [14] | 32.3 |
| BERT-S (RoBERTa-F) | 39.2 |
| METEOR [13] | 41.8 |
| CIDEr [15] | 43.9 |
| SPICE [16] | 44.9 |
| LEIC ($\tau_b$) [46] | 46.6 |
| BERT-S++ [32] | 46.7 |
| TIGEr [17] | 49.3 |
| NUBIA [47] | 49.5 |
| ViLBERTScore-F [18] | 50.1 |
| CLIP-S [19] | 51.2 |
| RefCLIP-S [19] | 53.0 |
| MID (ours) | **54.9** |

Table 5: Flickr8K-CF human judgment correlation.

| Method | Kendall $\tau_b$ |
|---|---|
| BLEU-4 [12] | 16.9 |
| CIDEr [15] | 24.6 |
| METEOR [13] | 22.2 |
| ROUGE-L [14] | 19.9 |
| SPICE [16] | 24.4 |
| BERT-S (RoBERTa-F) | 22.8 |
| LEIC [46] | 29.5 |
| CLIP-S [19] | 34.4 |
| RefCLIP-S [19] | 36.4 |
| MID (ours) | **37.3** |

**Comparison with the state-of-the-art models using MID.** Table 9 in Appendix E exhibits the MID performance of the recent text-to-image generative models along with the other major metrics.

## 4.2 Evaluation on image captioning

**Implementation details.** For a fair comparison with the current state-of-the-art evaluation metric (RefCLIP-S), we use the same pre-trained CLIP (ViT-B/32) used in the prior work to extract image and caption embedding vectors. We use the images and the corresponding reference captions to build the covariance matrices $\Sigma_{\mathbf{x}}, \Sigma_{\mathbf{y}}$ and the joint covariance matrix $\Sigma_{\mathbf{z}}$. For the numerical stability of the inverse of covariance matrix, we replace $\Sigma_{\mathbf{x}}^{-1}$ with $\tilde{\Sigma}_{\mathbf{x}}^{-1} = (\Sigma_{\mathbf{x}} + \epsilon\mathbb{I})^{-1}$, which handles the near-zero eigenvalues of covariance. We found that $\epsilon$ of 5e-4 generally works across all benchmark evaluations, except for the FOIL benchmark where we used $\epsilon$ of 1e-15, which was slightly better. Note that we use an identical prompt "A photo depicts" for all caption embeddings as employed in RefCLIP-S [19].

**Flickr8K-Expert and Flickr8k-CF.** We measure the correlation of the proposed method with the Likert-scale judgments, which indicate the relative correctness of given captions. Fliker8K-Expert [48] provides 17K human expert judgments for 5,664 images with a four-scale where the higher is better. Following prior works, we flatten all human judgments to a list of 16,992 (5,664×3) samples, and we exclude 158 pairs where their captions appear in the reference set. Flickr8K-CF [48] has 145K binary judgments from the CrowdFlower for 48K image-caption pairs. Each pair receives at least three judgments, and we take the proportion of positive as a corresponding score. Kendall's $\tau$ coefficient [42] measures the rank correlation, and $\tau_c$ and $\tau_b$ are used for Fliker8K-Expert and Flickr8K-CF, respectively. Although $\tau_c$ is more suitable when the underlying scales differ in two variables, we follow the previous works for a fair comparison. Tables 4 and 5 show the evaluation results. For both cases, our MID significantly improves the correlation of human judgments with

Table 6: Pascal-50S accuracy. Please see the text for the definition of subsets, HC, HI, HM, and MM.

| Method | HC | HI | HM | MM | Mean |
|---|---|---|---|---|---|
| length | 51.7 | 52.3 | 63.6 | 49.6 | 54.3 |
| BLEU-4 [12] | 60.4 | 90.6 | 84.9 | 54.7 | 72.6 |
| SPICE [16] | 63.6 | 96.3 | 86.7 | 68.3 | 78.7 |
| METEOR [13] | 63.8 | 97.7 | 93.7 | 65.4 | 80.1 |
| ROUGE-L [14] | 63.7 | 95.3 | 92.3 | 61.2 | 78.1 |
| CIDEr [15] | 65.1 | 98.1 | 90.5 | 64.8 | 79.6 |
| BERT-S (RoBERTa-F) | 65.4 | 96.2 | 93.3 | 61.4 | 79.1 |
| TIGEr [17] | 56.0 | **99.8** | 92.8 | 74.2 | 80.7 |
| ViLBERTScore-F [18] | 49.9 | 99.6 | 93.1 | 75.8 | 79.6 |
| BERT-S++ [32] | 65.4 | 98.1 | 96.4 | 60.3 | 80.1 |
| CLIP-S [19] | 56.5 | 99.3 | 96.4 | 70.4 | 80.7 |
| RefCLIP-S [19] | 64.5 | 99.6 | 95.4 | 72.8 | 83.1 |
| MID (ours) | **67.0** | 99.7 | **97.4** | **76.8** | **85.2** |

Table 7: FOIL hallucination pair-wise detection accuracy results. The methods utilize either one or four references.

| Method | 1-ref | 4-ref |
|---|---|---|
| length | 50.2 | 50.2 |
| BLEU-4 [12] | 66.5 | 82.6 |
| METEOR [13] | 78.8 | 85.4 |
| ROUGE-L [14] | 71.7 | 79.3 |
| CIDEr [15] | 82.5 | 90.6 |
| SPICE [16] | 75.5 | 86.1 |
| BERT-S | 88.6 | 92.1 |
| CLIP-S [19] | 87.2 | 87.2 |
| RefCLIP-S [19] | **91.0** | **92.6** |
| MID (ours) | 90.5 | 90.5 |

54.9 and 37.3, respectively. Notably, MID improves further than RefCLIP-S, which uses the same vision-and-language pre-trained CLIP.

**Flickr8K reference parsimony.** Compared with the other methods, our method does not directly rely on the corresponding references, but through the mean and covariance. Therefore, we could exploit the sample statistics with a limited number of references. Figure 4 shows the Kendall's $\tau$ correlation utilizing a subset of the available references. Interestingly, even though 30-40% of images are available, it retains the majority of performance. The dashed lines indicate the correlations of RefCLIP-S, which exploits all references. Notice that our method can be positioned between the reference-with-image and reference-free metrics. Because a sufficient amount of samples are required to assess image captioning models, the sample statistics can be reliable from the sufficient samples.

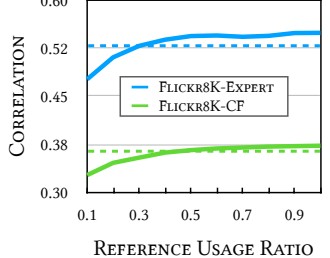

Figure 4: Flickr8K parsimony.

**Pascal-50S.** For a different evaluation setting of accuracy, the Pascal-50S [15] offers 4K pair-wise preference judgments between two captions, evenly splitting four categories; both are correct captions (HC), both human written, but one is incorrect (HI), one is from human, the other is by a model (HM), and both are generated by machine (MM). For each pair, there are 48 human judgments and the majority of votes decides which caption is preferred, where ties are broken randomly. As in the previous work [19], we randomly sample 5 references among 48 candidates and average over five evaluations. Table 6 shows the consistent results outperforming competitive methods. Except for HI, we achieve the state-of-the-art while the margin of HI is 0.1 having a near-perfect score of 99.7.

**Object hallucination sensitivity.** Rohrbach et al. [27] argue that image captioning models prone to generate the objects not presented in the image due to learned bias. To assess this aspect, the FOIL-COCO [40] builds the carefully modified captions from the COCO captions [2] by swapping a single noun-phrase, *e.g.*, substituting "cat" for "dog". To measure the accuracy whether assigning a higher score to the ground-truth caption over the FOIL caption, we evaluate 32K test images with exclusive four reference captions of the COCO dataset. Table 7 shows the competitive scores of our method. Since RefCLIP-S [19] directly accesses the reference captions of the evaluating image, they can exploit the original words (before foiling) in the references, which is roughly 87% in four references and 67% in a randomly selected reference. The RefCLIP-S is defined as follows:

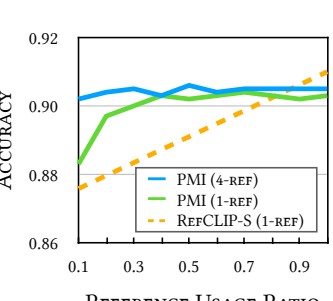

Figure 5: FOIL parsimony.

$$\text{RefCLIP-S}(\mathbf{x}, \mathbf{y}, \mathbf{R}) = H\big(\text{CLIP-S}(\mathbf{x}, \mathbf{y}), \max(\max_{\mathbf{r} \in \mathbf{R}} \cos(\mathbf{r}, \mathbf{y}), 0)\big) \tag{8}$$

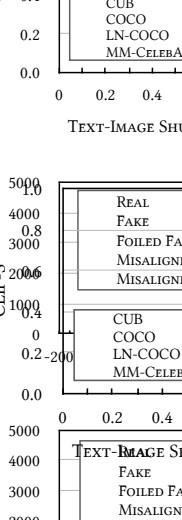

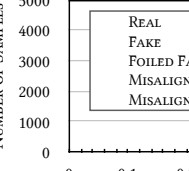

where $H$ is harmonic mean, $\mathbf{x}$ and $\mathbf{y}$ denotes image and caption embeddings, and $\mathbf{R}$ is a set of the references. Figure 5 supports this. Our method is robust via covariance estimation in both 1-ref (green line) and 4-ref (blue line), while RefCLIP-S is reducing to its reference-free version of CLIP-S degrading performance. We interpolate the scores of RefCLIP-S and CLIP-S as a reference (yellow line). If we similarly utilize the term of $\max(\max_{\mathbf{r} \in \mathbf{R}} \cos(\mathbf{r}, \mathbf{y}), 0)$ as in Equation 8, MID gets 92.4 and 93.7, outperforming RefCLIP-S (Appendix G for the detail, Figure 11 for the visualization).

## 5 General discussion

**Reliance on the pre-trained CLIP models.** Do the CLIP-based models take advantage of overfitting to the metric comparing other models? In Section 4.1.1, we *partly* answer that. Notice that multiple versions of the pre-trained CLIP vary the architectural type of visual backbone networks. We found that the normalized scores are generally consistent across generative models and the backbones, especially for our metric (Figure 3). However, it is worth noting that we encourage to use of the feature extractor (*e.g.*, CLIP) trained by the loss maximizing mutual information of two modalities (*ref.* the last paragraph of Section 3.1). Whether two independently-trained encoders perform well is under-explored; however, since we define the joint distribution in the proposed metric using the multivariate Gaussian distribution of concatenated features, we expect it reasonably works.

**Relations to the FID and CLIP-score.** The Fréchet inception distance (FID) assesses the perceptual quality of fake images from a generative model [5]. It is a symmetric distance where two differences between two means and two variances are independently considered and summed. While our method is an asymmetric divergence (Equation 7), the distances of means and variances are re-scaled to have unit variance as in the Mahalanobis distance (Equation 6), which is the source of domain robustness (Figure 1 Right). Note that the CLIP-S is the cosine similarity using CLIP features. The CLIP features are trained to minimize the InfoNCE loss by increasing mutual information approximately; however, our experiment shows that their approximation significantly degrades the evaluating performance.

**Measurement on diversity.** Our method does not entirely ignore the diversity by considering the covariance matrices from real and fake samples, similarly to the FID. However, a few unique fake samples (*e.g.*, mode collapse) may have small errors in the estimated statistics.

**Ethical considerations.** Multiple reports show that the CLIP and the generative models based on this pre-trained CLIP have problematic social biases toward racial or ethnic groups [31, 49]. Unless appropriate measures toward these biases in the CLIP, our metric is potentially subject to the risk of fairness, accountability, and transparency.

## 6 Conclusion

We, to our best knowledge, firstly argue that the negative cross-mutual information with multivariate Gaussian distributions can be used as a unified metric for multimodal generative models. We provide the theoretical analyses of the proposed metric by out-of-distribution detecting by the squared Mahalanobis distances, bias and variance decomposition, and the relation to the Kullback-Leibler divergence, along with the empirical experiments. We achieve the state-of-the-art performances on text-to-image generation and image captioning benchmarks, the generated and human Likert-scale judgment correlations, visual reasoning accuracy, Flickr8K-Expert, Flickr8K-CF, Pascal-50S, and FOIL hallucination detection. We look forward to seeing future works on the Gaussian cross-mutual information in multimodal representation learning based on this novel proposition.

## Acknowledgments

We sincerely thank Dongyoon Han for reviewing our manuscript and providing helpful comments. Also, we give thanks to Jung-Woo Ha for early discussions and suggestions throughout the project. And, we thank anonymous reviewers for their constructive feedback to solidify the manuscript. The NAVER Smart Machine Learning (NSML) platform [50] has been used in the experiments.

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
