# A Proofs

## A.1 Proof of the mutual information with Gaussian distributions

The mutual information of two Gaussian distributions is defined as:

$$I(\mathbf{X}; \mathbf{Y}) = \frac{1}{2} \log \left( \frac{\det(\Sigma_x) \det(\Sigma_y)}{\det(\Sigma_z)} \right). \tag{9}$$

*Proof.* Let the mutual information be:

$$I(\mathbf{X}; \mathbf{Y}) = \mathbb{E}_{p(\mathbf{x},\mathbf{y})} \left[ \log \frac{p(\mathbf{x}, \mathbf{y})}{p(\mathbf{x})p(\mathbf{y})} \right]. \tag{10}$$

Using the definition of multivariate Gaussian distribution as follows,

$$p(\mathbf{x}) = \frac{1}{\sqrt{(2\pi)^D \det(\Sigma)}} \exp\left[ -\frac{1}{2}(\mathbf{x} - \mu)^{\mathsf{T}} \Sigma^{-1} (\mathbf{x} - \mu) \right] \tag{11}$$

we rearrange the equation to cancel out the constant terms. Then, the continuous mutual information is reduced to:

$$I(\mathbf{X}; \mathbf{Y}) = \frac{1}{2} \log \left( \frac{\det(\Sigma_x) \det(\Sigma_y)}{\det(\Sigma)} \right) + \frac{1}{2} \mathbb{E}_{p(\mathbf{x},\mathbf{y})} \left[ D_M^2(\mathbf{x}) + D_M^2(\mathbf{y}) - D_M^2(\mathbf{z}) \right] \tag{12}$$

where $D_M^2$ denotes the squared Mahalanobis distance defined by $D_M^2(\mathbf{x}) = (\mathbf{x} - \mu_x)^{\mathsf{T}} \Sigma_{\mathbf{x}}^{-1} (\mathbf{x} - \mu_x)$, where $\mu_x$ and $\Sigma_{\mathbf{x}}$ are the mean and covariance of $\mathbf{x}$, and $\mathbf{z}$ denotes $[\mathbf{x}; \mathbf{y}]$. Notice that the squared Mahalanobis distance is one of the effective methods to detect anomalies in an unsupervised way [51].

By the way, the expectation of the squared Mahalanobis distance is the dimension of samples, $D$.

$$\mathbb{E}_{p(\mathbf{x})} D_M^2(\mathbf{x}) = \frac{1}{N} \text{tr}\left( \mathbf{X}^{\mathsf{T}} \Sigma_{\mathbf{x}}^{-1} \mathbf{X} \right) = \frac{1}{N} \text{tr}\left( \Sigma_{\mathbf{x}}^{-1} \mathbf{X} \mathbf{X}^{\mathsf{T}} \right) = \text{tr}\left( \Sigma_{\mathbf{x}}^{-1} \Sigma_{\mathbf{x}} \right) = \text{tr}(\mathbb{I}_D) = D \tag{13}$$

where $\mathbf{X} \in \mathbb{R}^{D \times N}$ is the samples, $\mathbb{I}_D \in \mathbb{R}^{D \times D}$ is the identity matrix. We use the cyclic property of trace where $\text{tr}(ABC) = \text{tr}(BCA)$. Therefore, the second term reduces to zero as follows:

$$\frac{1}{2} \mathbb{E}_{p(\mathbf{x},\mathbf{y})} \left[ D_M^2(\mathbf{x}) + D_M^2(\mathbf{y}) - D_M^2(\mathbf{z}) \right] = \frac{1}{2}(D + D - 2D) = 0. \tag{14}$$

We conclude the proof. □

By the way, the *point-wise mutual information (PMI)* with Gaussian distributions can be derived from Equation 12:

$$\text{PMI}(\mathbf{x}; \mathbf{y}) = I(\mathbf{X}; \mathbf{Y}) + \frac{1}{2} \left( D_M^2(\mathbf{x}) + D_M^2(\mathbf{y}) - D_M^2(\mathbf{z}) \right). \tag{15}$$

## A.2 The bias-variance decomposition of the expectation of the squared Mahalanobis distance

The expectation of PMI with respect to evaluating samples needs to calculate the expectation of three terms of the squared Mahalanobis distances (SMD) with respect to the evaluating sample $\hat{\mathbf{x}}$. With a notation of $\hat{\mathbf{X}} \in \mathbb{R}^{D \times N}$ for N evaluation samples, we can decompose the expectation of SMD with two terms of bias and variance as follows:

$$\mathbb{E}_{\hat{\mathbf{x}}} \left[ D_M^2(\hat{\mathbf{x}}) \right] = \frac{1}{N} \text{tr}\left( (\hat{\mathbf{X}} - \mu_{\mathbf{x}} \mathbb{1}^{\mathsf{T}})^{\mathsf{T}} \Sigma_{\mathbf{x}}^{-1} (\hat{\mathbf{X}} - \mu_{\mathbf{x}} \mathbb{1}^{\mathsf{T}}) \right) \tag{16}$$

$$= \frac{1}{N} \text{tr}\left( \Sigma_{\mathbf{x}}^{-1} (\hat{\mathbf{X}} - \mu_{\mathbf{x}} \mathbb{1}^{\mathsf{T}})(\hat{\mathbf{X}} - \mu_{\mathbf{x}} \mathbb{1}^{\mathsf{T}})^{\mathsf{T}} \right) \tag{17}$$

$$= \frac{1}{N} \text{tr}\left( \Sigma_{\mathbf{x}}^{-1} \left( \hat{\mathbf{X}} \hat{\mathbf{X}}^{\mathsf{T}} - N \mu_{\hat{\mathbf{x}}} \mu_{\hat{\mathbf{x}}}^{\mathsf{T}} + N(\mu_{\hat{\mathbf{x}}} - \mu_{\mathbf{x}})(\mu_{\hat{\mathbf{x}}} - \mu_{\mathbf{x}})^{\mathsf{T}} \right) \right) \tag{18}$$

$$= \text{tr}\left( \Sigma_{\mathbf{x}}^{-1} \left( \Sigma_{\hat{\mathbf{x}}} + (\mu_{\hat{\mathbf{x}}} - \mu_{\mathbf{x}})(\mu_{\hat{\mathbf{x}}} - \mu_{\mathbf{x}})^{\mathsf{T}} \right) \right) \tag{19}$$

$$= (\mu_{\hat{\mathbf{x}}} - \mu_{\mathbf{x}})^{\mathsf{T}} \Sigma_{\mathbf{x}}^{-1} (\mu_{\hat{\mathbf{x}}} - \mu_{\mathbf{x}}) + \text{tr}(\Sigma_{\mathbf{x}}^{-1} \Sigma_{\hat{\mathbf{x}}}) \tag{20}$$

$$= (\mu_{\hat{\mathbf{x}}} - \mu_{\mathbf{x}})^{\mathsf{T}} \Sigma_{\mathbf{x}}^{-1} (\mu_{\hat{\mathbf{x}}} - \mu_{\mathbf{x}}) + \text{tr}(\Sigma_{\mathbf{x}}^{-1} \Sigma_{\hat{\mathbf{x}}}) - \text{tr}(\Sigma_{\mathbf{x}}^{-1} \Sigma_{\mathbf{x}}) + \text{tr}(\Sigma_{\mathbf{x}}^{-1} \Sigma_{\mathbf{x}}) \tag{21}$$

$$= (\mu_{\hat{\mathbf{x}}} - \mu_{\mathbf{x}})^{\mathsf{T}} \Sigma_{\mathbf{x}}^{-1} (\mu_{\hat{\mathbf{x}}} - \mu_{\mathbf{x}}) + \text{tr}(\Sigma_{\mathbf{x}}^{-1} (\Sigma_{\hat{\mathbf{x}}} - \Sigma_{\mathbf{x}})) + D \tag{22}$$

where $\mathbb{1} \in \mathbb{R}^N$ is a vector of ones. Remind that the expectation of SMD is $D$ when the evaluating samples $\hat{x}$ are following the distribution of $x$ in Equation 13. However, the above equation shows that if the mean or covariance of $\hat{x}$ deviates from $x$, the result may be smaller or larger than D.

### A.3 Relation to Kullback–Leibler divergence

The proposed method MID is related to Kullback-Leibler divergence (or relative entropy). Let $\mathcal{N}_0(\mu_0, \Sigma_0)$ and $\mathcal{N}_1(\mu_1, \Sigma_1)$ are two multivariate normal distributions having the same dimension of $D$, then the Kullback-Leibler divergence between the distributions is as follows [52]:

$$D_{\mathrm{KL}}\left(\mathcal{N}_0 \parallel \mathcal{N}_1\right) = \frac{1}{2}\left(\mathrm{tr}\left(\Sigma_1^{-1}(\Sigma_0 - \Sigma_1)\right) + (\mu_1 - \mu_0)^\mathsf{T}\Sigma_1^{-1}(\mu_1 - \mu_0) + \log\left(\frac{\det\Sigma_1}{\det\Sigma_0}\right)\right).$$

Using the above equation and Equation 22, we rearrange Equation 12 as follows:

$$\begin{aligned}
\mathbb{E}_{(\hat{x},\hat{y})\sim\mathcal{D}}\mathrm{PMI}(\hat{x};\hat{y}) =& I(\mathbf{X};\mathbf{Y}) + D_{\mathrm{KL}}(p(\hat{x}) \parallel p(\mathbf{x})) + D_{\mathrm{KL}}(p(\hat{y}) \parallel p(\mathbf{y})) - D_{\mathrm{KL}}(p(\hat{z}) \parallel p(\mathbf{z})) \\
& - \frac{1}{2}\left(\log\left(\frac{\det\Sigma_{\mathbf{x}}}{\det\Sigma_{\hat{x}}}\right) + \log\left(\frac{\det\Sigma_{\mathbf{y}}}{\det\Sigma_{\hat{y}}}\right) - \log\left(\frac{\det\Sigma_{\mathbf{z}}}{\det\Sigma_{\hat{z}}}\right)\right) \quad (23) \\
=& I(\mathbf{X};\mathbf{Y}) + D_{\mathrm{KL}}(p(\hat{x}) \parallel p(\mathbf{x})) + D_{\mathrm{KL}}(p(\hat{y}) \parallel p(\mathbf{y})) - D_{\mathrm{KL}}(p(\hat{z}) \parallel p(\mathbf{z})) \\
& - \frac{1}{2}\log\left(\frac{\det\Sigma_{\mathbf{x}}\det\Sigma_{\mathbf{y}}}{\det\Sigma_{\mathbf{z}}}\right) + \frac{1}{2}\log\left(\frac{\det\Sigma_{\hat{x}}\det\Sigma_{\hat{y}}}{\det\Sigma_{\hat{z}}}\right) \quad (24) \\
=& I(\hat{\mathbf{X}};\hat{\mathbf{Y}}) + D_{\mathrm{KL}}(p(\hat{x}) \parallel p(\mathbf{x})) - D_{\mathrm{KL}}(p(\hat{z}) \parallel p(\mathbf{z})) \quad (25)
\end{aligned}$$

where $D_{\mathrm{KL}}(p(\hat{y}) \parallel p(\mathbf{y})) = 0$ since $\hat{y}$ and $y$ are the same condition evaluating generations.

## B   Generated Likert-scale judgment correlation using VQ-Diffusion

Table 8 and Figure 6 show the results from the (foiled) fake images using VQ-Diffusion [43]. While the proposed MID outperforms the competing methods, the portion of fake images that get higher scores than real images is decreased in InfoNCE and CLIP-S. This observation may attribute to the under-performance of VQ-Diffusion than LAFTIE or the side effect of the contrastive loss used in LAFITE. Remind that our method shows consistency toward different models among the comparative metrics.

Table 8: Generated Likert-scale judgment correlation using VQ-Diffusion. † uses the i.i.d. samples having at least one detected object, which was 88.1%, to calculate the SOA accuracy per image.

| Method | Backbone | Kendall $\tau_c$ | Kendall $\tau_b$ |
|---|---|---|---|
| SOA† [8] | YOLO-V3 | 37.0 | 38.4 |
| CLIP-S [19] | CLIP (ViT-B/32) | 70.3 | 60.9 |
| InfoNCE [38] | CLIP (ViT-B/32) | 74.2 | 64.3 |
| CLIP-R-Precision [11] | CLIP (ViT-B/32) | 66.5 | 54.5 |
| OFA-Captioning+CLIP-S [19, 39] | OFA-Large + CLIP (ViT-B/32) | 73.8 | 63.9 |
| CLIP-S [19] | CLIP (ViT-L/14) | 70.9 | 61.4 |
| InfoNCE [38] | CLIP (ViT-L/14) | 78.0 | 67.6 |
| CLIP-R-Precision [11] | CLIP (ViT-L/14) | 68.5 | 56.5 |
| OFA-Captioning+CLIP-S [19, 39] | OFA-Large + CLIP (ViT-L/14) | 74.2 | 64.3 |
| MID (ours) | CLIP (ViT-B/32) | 79.8 | 69.1 |
| MID (ours) | CLIP (ViT-L/14) | **82.0** | **71.1** |

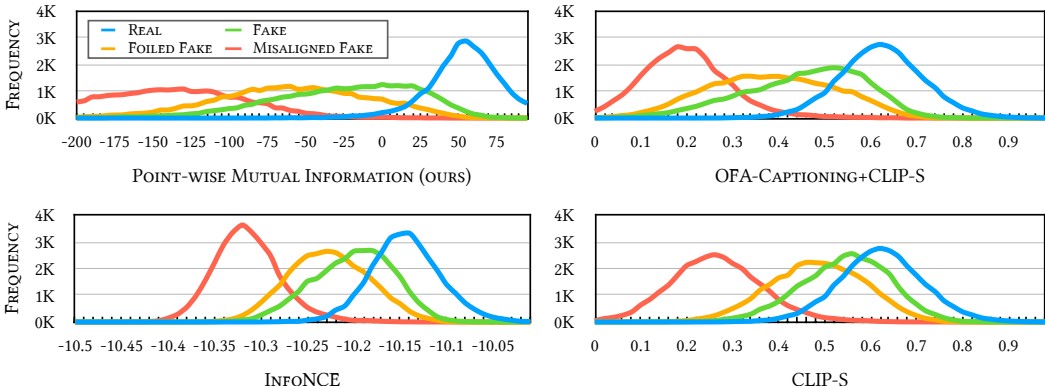

Figure 6: The histogram shows the frequencies of the competing methods using the CLIP of ViT-L/14 for the four-scale judgments from the COCO dataset. The real images and the original captions (**Real**), the fake images generated from the VQ-Diffusion [43] using the original caption (**Fake**), the fake images generated from the VQ-Diffusion but using the foiled caption deliberately swapping some objects to the other words (**Foiled Fake**) (notice that the foiled fake images are still matching with the original captions to assess), and randomly matching the foiled fake to the original captions by shuffling (**Misaligned Fake**). For the detail of methods, please see the text in Section 4.

## C   The details on visual reasoning accuracy

We describe the detail of visual reasoning accuracy in Table 3. For the object task, we use randomly sampled 30K captions from the FOIL dataset [40]. For the count task, we use a set of tokens "0", "1", "2", "3", "4", "one", "two", "three", and "four". For the color task, we use the sixteen basic color keywords [5]. For the spatial relationship task, we use "above", "below", "left", "right", "front", and "back". The numbers of samples are 30K, 1.3K, 4.6K, 1.5K for the object, count, color, and spatial relationship tasks, respectively. The LAFITE [41] generates the (foiled) fake images.

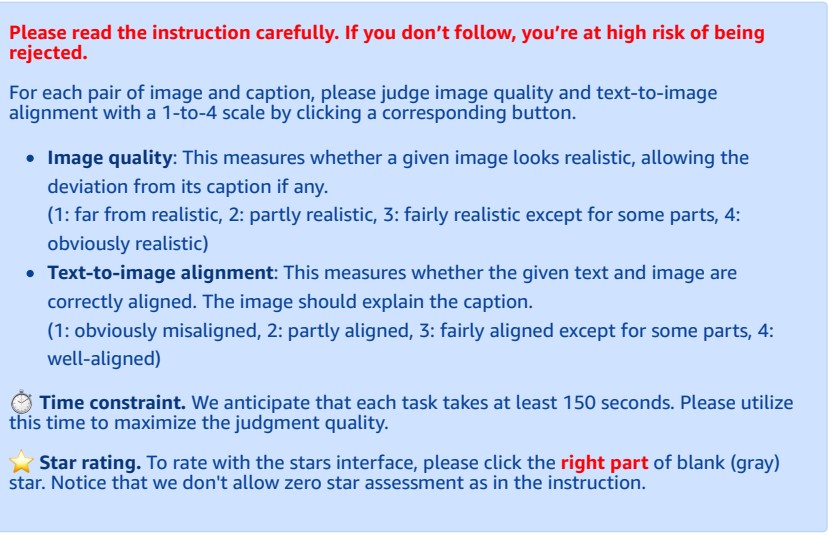

Figure 7: The instruction appeared on top of the AMT task interface.

---

[5] https://www.w3.org/TR/css-color-3/#html4

1

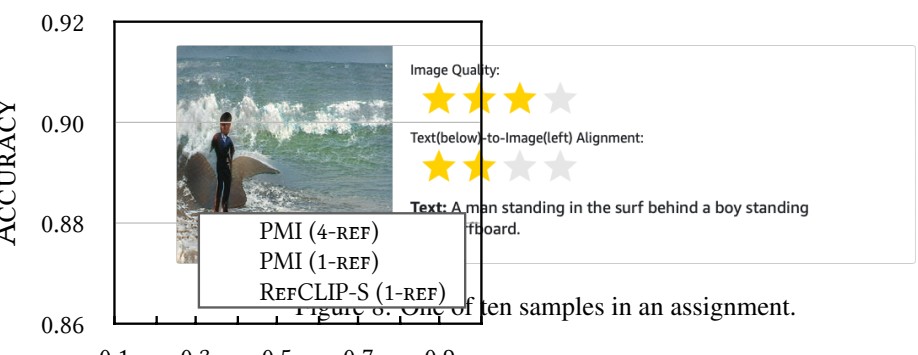

Figure 8: One of ten samples in an assignment.

## D Human judgment collection from the Amazon Mechanical Turk

In Section 4.1, the human Likert-scale judgments on the fine-grained text-image alignment are used to evaluate the metrics and reported the rank correlations in Table 2. The following paragraphs describe the data collection procedure using the Amazon Mechanical Turk (AMT).

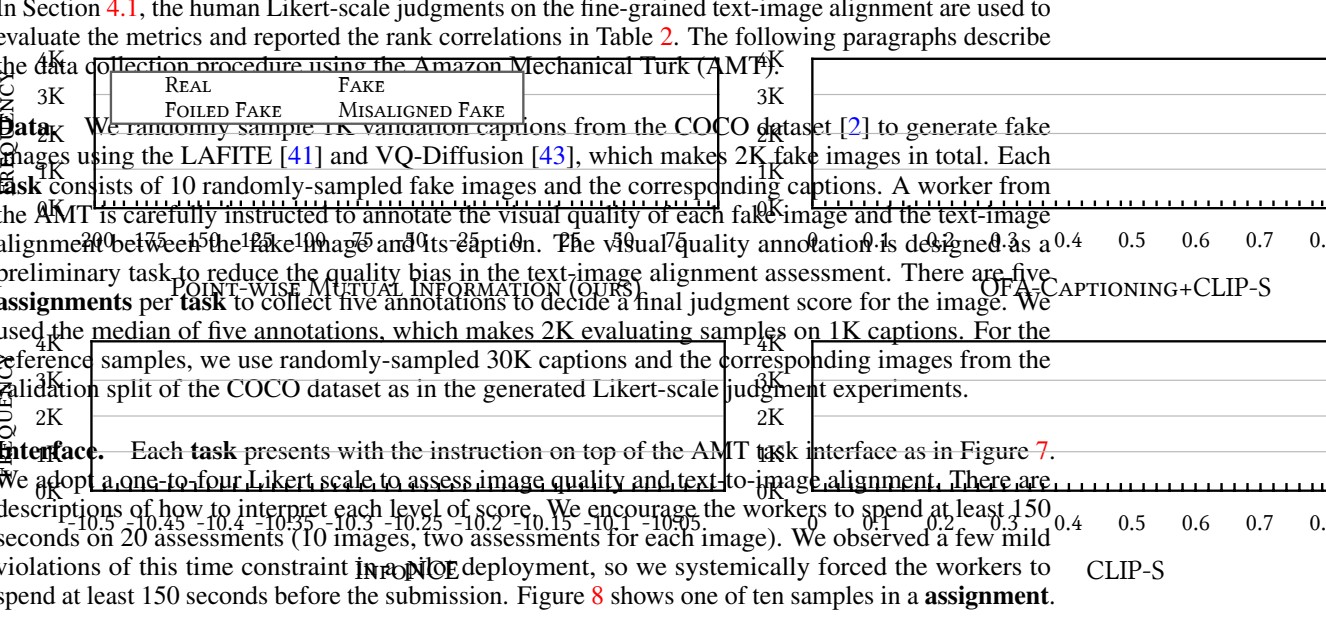

**Data.** We randomly sample 1K validation captions from the COCO dataset [2] to generate fake images using the LAFITE [41] and VQ-Diffusion [43], which makes 2K fake images in total. Each **task** consists of 10 randomly-sampled fake images and the corresponding captions. A worker from the AMT is carefully instructed to annotate the visual quality of each fake image and the text-image alignment between the fake image and its caption. The visual quality annotation is designed as a preliminary task to reduce the quality bias in the text-image alignment assessment. There are five **assignments** per **task** to collect five annotations to decide a final judgment score for the image. We used the median of five annotations, which makes 2K evaluating samples on 1K captions. For the reference samples, we use randomly-sampled 30K captions and the corresponding images from the validation split of the COCO dataset as in the generated Likert-scale judgment experiments.

**Interface.** Each **task** presents with the instruction on top of the AMT task interface as in Figure 7. We adopt a one-to-four Likert scale to assess image quality and text-to-image alignment. There are descriptions of how to interpret each level of score. We encourage the workers to spend at least 150 seconds on 20 assessments (10 images, two assessments for each image). We observed a few mild violations of this time constraint in our deployment, so we systemically forced the workers to spend at least 150 seconds before the submission. Figure 8 shows one of ten samples in a **assignment**.

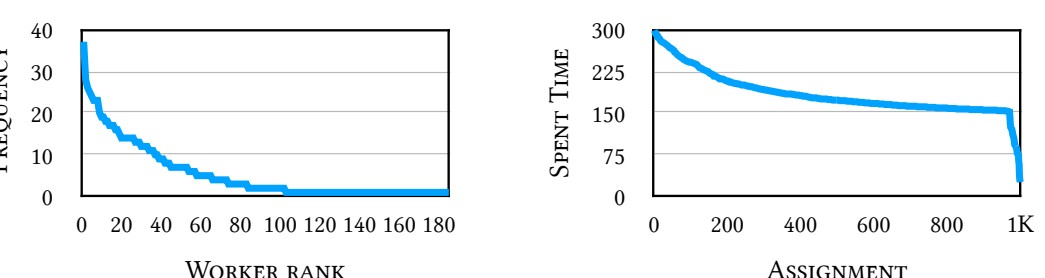

Figure 9: **Left.** The number of assignments per worker. **Right.** Time spent per assignment.

**Cost.** We collected 1K **assignments** for 200 **tasks**. Considering reasonable earn per hour, we paid $0.21 for each assignment, $210 in total.

**Statistics.** Figure 9 shows the number of assignments per worker (left) and the time spent per assignment. A worker did at most 37 assignments and the assignments are done within five minutes. After taking the median, the mean and standard deviation of the quality judgments are 2.43 and 0.61, respectively, and these of the alignment judgments are 2.62 and 0.64, respectively.

**Visualization** Figure 10 shows some examples comparing with human judgment scores. Note that we used the mean of three median annotations from workers for this visualization.

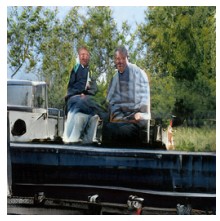

**Two men sitting on the rails of a truck.**
*LAFITE*

CLIP-S: 0.73 (0.89)
MID (ours): **0.79** (44.32)
Human: 0.91 (3.33)

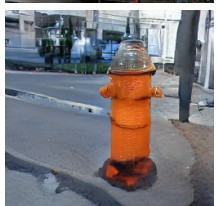

**Orange fire hydrant on the sidewalk in a commercial area.**
*LAFITE*

CLIP-S: 0.07 (0.60)
MID (ours): **0.09** (-28.34)
Human: 0.31 (2.33)

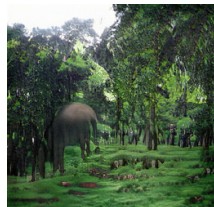

**An elephant is in the distance of a green area with many large leafy trees.**
*LAFITE*

CLIP-S: 0.13 (0.64)
MID (ours): **0.39** (16.87)
Human: 0.53 (2.67)

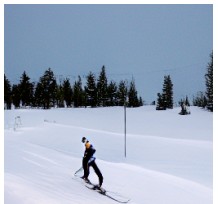

**A guy on skis going down a snow covered hill between trees.**
*VQ-Diffusion*

CLIP-S: 0.12 (0.64)
MID (ours): **0.30** (8.10)
Human: 0.75 (3.00)

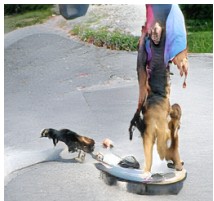

**A person riding a skate board with a dog following beside.**
*LAFITE*

CLIP-S: 0.99 (1.08)
MID (ours): **0.57** (30.66)
Human: 0.53 (2.67)

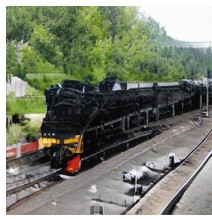

**A large black train on a train track.**
*LAFITE*

CLIP-S: 0.48 (0.79)
MID (ours): **0.67** (36.36)
Human: 0.75 (3.00)

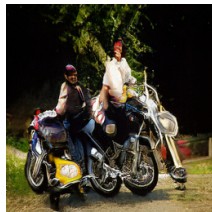

**A couple ride happily on their motorcycle.**
*LAFITE*

CLIP-S: 0.29 (0.72)
MID (ours): **0.53** (28.02)
Human: 0.53 (2.67)

Figure 10: The visualization of generated images and their evaluation results. From the top, a caption for generation, a generative model type, a normalized rank, and a raw score in parentheses for each metric. The normalized rank denotes the evaluated score rank divided by the number of samples (zero-to-one scale) for a fair comparison. **bold** indicates the closer to human judgments.

# E Performance results of text-to-image generative models

Table 9: The performance reports of text-to-image generative models. [†]The publicly-released model using the filtered subset of training dataset (https://github.com/openai/glide-text2im). For the backbone, I, Y, C stands for Inception-V3 [7], YOLO-V3 [26], and CLIP [10].

| Metric
Backbone | FID ↓
I | SOA-C ↑
Y | SOA-I ↑
Y | FID (ours) ↓
I | MID ↑
C (ViT-B/32) | MID ↑
C (ViT-L/14) |
|---|---|---|---|---|---|---|
| GLIDE [53][†] | - | - | - | 32.08±.05 | 1.00±.16 | 1.03±.06 |
| AttnGAN [24] | 33.10±.11 | 25.88 | 39.01 | 29.15±.06 | -8.90±.18 | -65.20±.92 |
| DM-GAN [45] | 27.34±.11 | 33.44 | 48.03 | 22.90±.06 | 3.51±.20 | -44.66±.71 |
| OP-GAN [8] | 24.70±.09 | 35.85 | 50.47 | 22.14±.01 | -1.32±.10 | -50.34±.84 |
| DF-GAN [54] | 21.42 | - | - | 31.75±.06 | -15.21±.12 | -58.75±.16 |
| VQ-Diffusion [43] | 13.86 | - | - | 13.13±.05 | 5.77±.11 | -19.63±.28 |
| LAFITE [41] | **8.12** | **61.09** | **74.78** | **8.03±.01** | **35.17±.20** | **6.26±.69** |
| Real | 6.09±.05 | 74.97 | 80.84 | 2.73±.15 | 41.63±.06 | 57.44±.06 |

Table 9 illustrates the performance of text-to-image generative models, including the proposed MID scores. The reports of FID, SOA-C, and SOA-I are from Hinz et al. [8] while missing scores are from the corresponding cites. They "randomly sampled three times 30,000 images from the training set and compared them to the statistics of the validation set." We found that LAFITE uses all 82,612 training images to get the statistics, so we also report FID (ours) for a fair comparison. We sampled 30,000 fake images from the validation captions. This different sampling strategy may attribute to the difference in the real images' upper bound (6.09 vs. 2.73). For the proposed MID, we randomly sampled 30,000 images and the captions for each image from the validation set as reference samples $(\mathbf{X}, \mathbf{Y})$. We trained the LAFITE model from scratch using the official code [6] with the same hyperparameters and 1.5 times training longer to achieve a slightly better FID than the publicly released model. Interestingly, the filtered GLIDE underperforms with the worst FID; however, it outperforms some of the other models with MID. It may show that the data filtering severely affects FID while relatively retaining the performance of text-image alignment captured by MID.

# F Performance comparison on the comp-t2i dataset

Table 10 demonstrates the human judgement correlation for the comp-t2i dataset [11]. We compute Pearson correlation coefficient (PCC), Spearman correlation coefficient (SCC), binary decision consensus accuracy (Acc.), and Kendall's $\tau$ coefficients between metric scores and human judgment scores. The human judgment scores are pre-processed using the ratio of $n/5$ where $n$ is the number of votes. Pearson correlation coefficient is a statistic that measures the linear correlation, while Spearman correlation coefficient evaluates a monotonic relationship rather than the raw values. Following [11], the accuracy is measured for binary decision consensus for the seen and swapped captions. Additionally, we report two variants of Kendall's $\tau$ having a better confidence interval for more precise comparison.

We clarify that even with the released asset, we cannot reproduce the reported scores for the CLIP-R-Precision of some splits. So, we report our reproduction along with their reported scores. Following the released code, we exploit available bounding boxes for the image pre-processing and sample the negative examples from a randomly-chosen different class for the CLIP-R-Precision. Notice that CLIP-R-Precision has a moderate variance since the sampled negative examples impact the score while risking some degree of false negative describing narrow domains of birds and flowers.

We conducted the experiment using both the pre-trained CLIP and the CLIP model further fine-tuned on the corresponding dataset. Note that CLIP model is based on ResNet 101. Since CLIP-R-Precision score is determined by a set of caption candidates, we repeat 10 times to construct the negative examples and report the standard deviation. To measure the proposed MID, the training set for fine-tuning CLIP is used as the reference set $(\mathbf{X}, \mathbf{Y})$ in Equation 25 (Appendix), while the scored image-caption pairs are the evaluation samples $(\hat{\mathbf{X}}, \hat{\mathbf{Y}})$. In most cases, regardless of whether it is

---

[6]https://github.com/drboog/Lafite

fine-tuned or not, the proposed MID outperforms CLIP-S and CLIP-R-Precision. These results suggest that our MID has better generalization capabilities.

Table 10: The comp-t2i human judgment correlation [11]. 'FT' denotes whether the pre-trained CLIP model is further fine-tuned on the corresponding dataset which is available at `https://github.com/Seth-Park/comp-t2i-dataset`, and † denotes our reproduction. The best result and the second best result are boldfaced and underlined, respectively.

| | Metric | FT | PCC | SCC | Acc. | $\tau_c$ | $\tau_b$ |
|---|---|---|---|---|---|---|---|
| **C-CUB Color** | Human [11] | - | 0.5949 | 0.5890 | 81.7 | - | - |
| | DAMSM [11, 24] | - | 0.0503 | 0.1224 | 54.9 | - | - |
| | CLIP-S [19] | ✗ | 0.1919 | 0.1865 | 67.0 | 14.78 | 13.59 |
| | CLIP-R-Precision [11]† | ✗ | 0.1410±.013 | 0.1426±.014 | 53.3±1.3 | 13.02±1.25 | 12.62±1.27 |
| | MID (ours) | ✗ | **0.2863** | **0.3499** | **68.5** | **27.68** | **25.45** |
| | CLIP-S [19] | ✓ | **0.3711** | 0.3428 | **77.5** | 27.24 | 25.05 |
| | CLIP-R-Precision [11] | ✓ | 0.0752 | 0.1263 | 56.4 | - | - |
| | CLIP-R-Precision [11]† | ✓ | 0.2538±.014 | 0.2474±.014 | 62.1±1.4 | 28.04±1.52 | 21.90±1.19 |
| | MID (ours) | ✓ | 0.3558 | **0.3990** | 76.5 | **31.80** | **29.24** |
| **C-CUB Shape** | Human [11] | - | 0.3949 | 0.4007 | 70.0 | - | - |
| | DAMSM [11, 24] | - | 0.1229 | 0.0170 | 52.7 | - | - |
| | CLIP-S [19] | ✗ | 0.0287 | 0.0315 | **61.0** | 2.37 | 2.17 |
| | CLIP-R-Precision [11]† | ✗ | 0.0071±.012 | 0.0078±.012 | 43.9±0.6 | 0.7±1.07 | 0.69±1.05 |
| | MID (ours) | ✗ | **0.1113** | **0.1079** | 55.0 | **8.56** | **7.85** |
| | CLIP-S [19] | ✓ | 0.0577 | 0.0593 | 56.5 | 4.64 | 4.26 |
| | CLIP-R-Precision [11] | ✓ | 0.0878 | 0.0806 | 52.2 | - | - |
| | CLIP-R-Precision [11]† | ✓ | 0.1096±.022 | 0.1063±.022 | 46.3±1.0 | **11.97±2.40** | **9.40±1.90** |
| | MID (ours) | ✓ | **0.1118** | **0.1280** | 60.0 | 9.96 | 9.14 |
| **C-Flower Color** | Human [11] | - | 0.5891 | 0.5870 | 80.5 | - | - |
| | DAMSM [11, 24] | - | -0.0457 | 0.0456 | 52.3 | - | - |
| | CLIP-S [19] | ✗ | 0.1802 | 0.1823 | **70.0** | 14.29 | 13.13 |
| | CLIP-R-Precision [11]† | ✗ | 0.0939±.011 | 0.0943±.012 | 49.5±0.5 | 8.95±1.13 | 8.34±1.04 |
| | MID (ours) | ✗ | **0.2872** | **0.3433** | 69.0 | **26.95** | **24.76** |
| | CLIP-S [19] | ✓ | **0.4486** | **0.4321** | **81.0** | **34.85** | **32.03** |
| | CLIP-R-Precision [11] | ✓ | 0.2818 | 0.3151 | 65.3 | - | - |
| | CLIP-R-Precision [11]† | ✓ | 0.2892±.014 | 0.2864±.014 | 58.8±1.2 | 31.04±1.51 | 25.33±1.25 |
| | MID (ours) | ✓ | 0.4156 | 0.4289 | 74.0 | 34.48 | 31.68 |
| **C-Flower Shape** | Human [11] | - | 0.3721 | 0.3698 | 68.8 | - | - |
| | DAMSM [11, 24] | - | -0.0435 | 0.0876 | 48.0 | - | - |
| | CLIP-S [19] | ✗ | 0.0330 | 0.0300 | 54.0 | 2.31 | 2.12 |
| | CLIP-R-Precision [11]† | ✗ | 0.0613±.016 | 0.0604±.016 | 46.4±1.8 | 5.99±1.65 | 5.34±1.45 |
| | MID (ours) | ✗ | **0.0893** | **0.0994** | **59.0** | **7.93** | **7.28** |
| | CLIP-S [19] | ✓ | 0.0875 | 0.0809 | 57.0 | 6.40 | 5.87 |
| | CLIP-R-Precision [11] | ✓ | 0.0920 | 0.0702 | 52.0 | - | - |
| | CLIP-R-Precision [11]† | ✓ | 0.0467±.017 | 0.0473±.017 | 46.2±1.0 | 5.38±1.92 | 4.18±1.49 |
| | MID (ours) | ✓ | **0.1056** | **0.1212** | **63.5** | **9.66** | **8.87** |

# G  Utilizing the references like RefCLIP-S for FOIL hallucination detection

When we similarly utilize the term of $\max(\max_{\mathbf{r} \in \mathbf{R}} \cos(\mathbf{r}, \mathbf{y}), 0)$ in Equation 8 for MID , it results in 92.4 and 93.7, outperforming RefCLIP-S. Since the scoring scale of MID is different from cosine similarity, we use the arithmetic mean instead of the harmonic mean and a different weight $\alpha$ of 3e2, considering the difference between their standard deviations as follows:

$$\text{RefMID}(\mathbf{x}, \mathbf{y}, \mathbf{R}) = \frac{1}{2}\big(\text{MID}(\mathbf{x}, \mathbf{y}), \alpha \max(\max_{\mathbf{r} \in \mathbf{R}} \cos(\mathbf{r}, \mathbf{y}), 0)\big) \tag{26}$$

where MID is parameterized by the referencing image and caption statistics. Notice that RefCLIP-S also has the hyper-parameter of 2.5 in the CLIP-S to balance with the cosine similarity term.

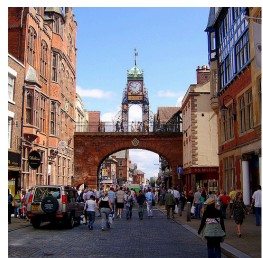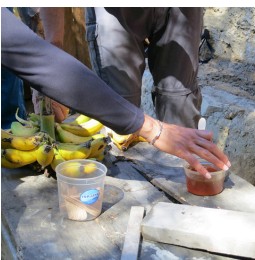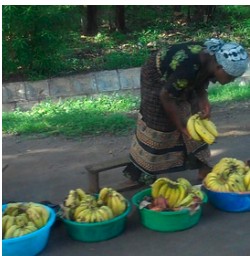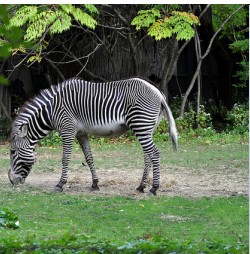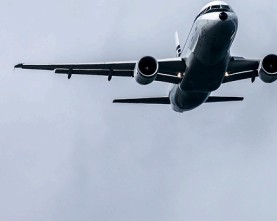

**FO**: A tall **book** tower with people walking down a city street. (CLIP-S/RefCLIP-S/ PMI=.737/.804/16.9)
**GT**: A tall **clock** tower with people walking down a city street. (.738/.806/20.0)

**FO**: Some people a **chair** some bananas and plastic cups. (.747/.832/8.63)
**GT**: Some people a **table** some bananas and plastic cups. (.756/.843/12.0)

**FO**: Large bowls of **broccoli** bunches being examined by a female buyer. (.774/.811/11.8)
**GT**: Large bowls of **banana** bunches being examined by a female buyer. (.761/.815/9.61)

**FO**: a **giraffe** grazing on grass in an open field. (.707/.747/-4.89)
**GT**: A **zebra** grazing on grass in an open field. (.718/.821/38.6)

**FO**: A **boat** that is flying in t sky. (.728/.791/12.7)
**GT:** A **airplane** that is flying the sky. (.779/.845/37.2)

Figure 11: The visualization of the FOIL hallucination detection. **FO** and **GT** denote FOIL and ground-truth captions, respectively. The foiled word is highlighted in red, and its counterpart is in blue. For each caption, we report the scores of CLIP-S [19], RefCLIP-S [19], and MID (ours). The first two columns show the corrected examples while the outlined column shows an example that CLIP-S and MID are failed to detect. The smaller score is better for the FOIL captions. As pointed out in Section 4.2, RefCLIP-S directly exploits the reference captions having the counterpart of the foiled word. One of the references of the third example was that "a woman is picking *bananas* from a basket." The fourth example shows that MID can be negative for unlikely samples since it is based on the definition of differential entropy.