# OpenReview forum: "Mutual Information Divergence: A Unified Metric for Multimodal Generative Models"
_NeurIPS.cc/2022/Conference — NeurIPS 2022 Accept_

### Official Review · Reviewer_wkz2 · 2022-07-03

**Rating:** 7
**Confidence:** 3
**Soundness:** 3 good
**Presentation:** 3 good
**Contribution:** 3 good

**Summary:**

This paper proposes a new metric for evaluating text-to-image and image-to-text models. The metric is mutual information divergence, which uses CLIP features as a source of ground truth. The authors provide theoretical analysis of their metric and demonstrate that it outperforms other metrics on numerous evaluations of real vs. fake images and captions.

**Questions:**

To what extent does CLIP’s training distribution play a role in the strong performance observed? If the model were scaled to 4 billion images, for example, would MID improve further?

**Limitations:**

The paper does not have a limitations section. It could use a discussion of the biases in CLIP. See, for example, Wolfe et al. 2022 ACM FAccT; Birhane et al. 2021 arXiv.

**Strengths And Weaknesses:**

Originality
Strengths: This is a novel area of inquiry and the metric employed achieves better results than any other proposed method.
Weaknesses: There have been methods that employed CLIP features previously, and the improvement over these is relatively small.

Quality
Strengths: Outperformance of other methods. Strong theoretical evaluation of the proposed method.
Weaknesses: No discussion of the potential limitations and biases of CLIP.

Clarity
Strengths: Visualizations are easy to understand. Intro and mathematical descriptions are well presented.
Weaknesses: I found the paper hard to follow at times, especially as it moves into evaluation and discussions.

Significance
Strengths: Outperformance of other methods. Wide variety of environments in which the method might be useful for making sense of the relationship between text and images.
Weaknesses: Many of the best synthetic generators and image captioners now directly employ the CLIP embedding space to achieve strong results (DALL-E models; GLIDE; VQGAN-CLIP; Antarctic Captions, CLIP prefix-LM and its cousins; etc.). Won’t using the CLIP embedding space as a means of evaluating multimodal representations be a confound whenever these systems that already employ a CLIP model need to be evaluated?

---

> ### Author Response · Authors · 2022-08-02
> **Author feedback for Reviewer wkz2**
>
> **1. Potential Limitations and Biases of CLIP.** We will add the dedicated section for the limitation as follows:
>
> > **Ethical considerations.** There are multiple reports that the CLIP or the generative models based on this pretrained CLIP has problematic social biases toward racial or ethnic groups (Cho et al., 2022, Wolfe et al., 2022). Unless appropriate measures toward these biases in the CLIP, our metric is potentially subject to the risk of fairness, accountability, and transparency.
>
> > **CLIP-reliant.** Do the CLIP-based models take advantage of overfitting to the metric comparing other models? Sec 4.1.1 “Inspecting possible over-fitting with the CLIP features” partly answers that. Multiple versions of the pretrained CLIP vary the architectural type of visual backbone networks. We found that the normalized scores are generally consistent across generative models and the backbones, especially for our metric (see Fig. 3). However, it is worth noting that we encourage to use of the feature extractor (e.g., CLIP) with the training loss maximizing mutual information of two modalities (See the last paragraph of Sec 3.1; L125-129). Since we expect efficient multimodal representation learning helps for accurate metrics for multimodal alignments.
>
> **2. If the model were scaled to 4 billion images, for example, would MID improve further?** Although it is hard to predict, we observed that the ViT-L/14 consistently outperforms ViT-B/32, expecting there is a room for scaliability, so the increase of data size may result in substantial performance gain.

---

> > ### Comment · Reviewer_wkz2 · 2022-08-09
> > **Response to author revisions**
> >
> > The author's response and revisions largely address my concerns. I think CLIP-reliance could still be an issue where a CLIP-like model is assessed, but given the array of potential applications of the method, it is okay to just address this as a limitation.

---

### Official Review · Reviewer_WuSm · 2022-07-08

**Rating:** 6
**Confidence:** 2
**Soundness:** 3 good
**Presentation:** 3 good
**Contribution:** 3 good

**Summary:**

This paper introduces the Mutual Information Divergence (MID) metric for multimodal generative models, where the metric attempts to measure the mutual information between conditions and generations under the Gaussian assumption. Various experiments on text-to-image generation / image captioning, and theoretical analysis are performed to demonstrate the effectiveness and rationality of the proposed metric. It is interesting that Human Likert-scale judgment is employed to validate the superiority of MID.

**Questions:**

(1)   Are there any analysis on the relation between MID and existing measuring metrics, such as FID, or image-text matching score?
(2)   Intuitively, MID impose higher alignment between image and text, does this hurt the diversity？

**Limitations:**

Please see questions and weaknesses.

**Strengths And Weaknesses:**

- Strengths
(1) This paper introduce the MID metric for multimodal generative models, which is somewhat interesting and brings more reasonable results.
(2) The paper conducts solid experiments to validate the effectiveness of MID on both text-to-image generation and image captioning tasks, such as the generated and human Liker-scale judgment correlations, visual reasoning accuracy, Flickr8K-Expert, Flickr8K-CF, Pascal-50S, and FOIL hallucination detection.

- Weaknesses
(1) It lacks necessary analysis on relations between MID and existing metrics listed in Related Work.

---

> ### Author Response · Authors · 2022-08-02
> **Author feedback for Reviewer WuSm**
>
> **1. Relation Analysis to FID and CLIP-Score.** We believe it is worth including in the main paper. We consider the following paragraph for Sec 3.4 (new):
>
> > The Fréchet inception distance (FID) assesses the perceptual quality of fake images from a generative model (Heusel et al., 2017). It is a symmetric distance where two differences between two means and two variances are independently considered and summed. While our method is an asymmetric divergence (Eqn.7), the distances of means and variances are rescaled to have unit variance as in the Mahalanobis distance (Eqn.6), which is the source of domain robustness (Fig.1 Right). Note that the CLIP-S is a cosine similarity distance using CLIP features. The CLIP features are trained to minimize the InfoNCE loss by increasing mutual information approximately; however, our experiment shows that their approximation significantly degrades the evaluating performance.
>
> **2. Diversity.** This is a good point whether this metric may neglect the diversity of generations. In short, our method does not entirely neglect the diversity by considering the covariance matrices from real and fake samples, similarily to the FID. However, it is possible that a few unique fake samples (e.g., mode collapse) may have a smaller error in the covariance matrix.

---

> > ### Comment · Reviewer_WuSm · 2022-08-10
> > **Final Rating**
> >
> > Most of my concerns have been addressed, and I will raise my vote to "weak accept".

---

### Official Review · Reviewer_adUF · 2022-07-10

**Rating:** 6
**Confidence:** 4
**Soundness:** 4 excellent
**Presentation:** 4 excellent
**Contribution:** 3 good

**Summary:**

This submission proposes a new evaluation metric for visual-language generation tasks. Compared to previous metrics, the authors propose to leverage negative cross-mutual information with multivariate Gaussian distributions to calculate mutual information. The authors conduct experiments on both text-to-image generation and image captioning datasets and show the effectiveness of the proposed metrics.

**Questions:**

Overall I am satisfied with the submission. There is one problem with the metric:

The title focuses on "Multimodal Generative Models". However this metric is limited only within visual and textual modalities. As shown in Figure 1 left, to calculate the metrics, it also needs to use CLIP model to extract embedding first. This may limit generalization of the metrics. In the experiments, the authors also focus on text-to-image and image captioning tasks. It would be better to show the usage of the metric on more "Multimodal Generation" tasks (e.g., Video, sound, point cloud, etc.) to increase its scope.

**Limitations:**

Please check the "Questions" section above.

**Strengths And Weaknesses:**

+ The motivation is clear and proposed framework is easy to understand. Besides, the authors also released the source code to reproduce the metric calculation.

+ The technical details are clearly explained with valid experiments' results support.

+ The authors also conduct ablation study to discuss the metrics usage on both text-to-image and image captioning tasks on various datasets, which is much appreciated.

---

> ### Author Response · Authors · 2022-08-02
> **Regarding "Multimodal Generative Models"**
>
> As *Reviewer Ezue* noticed, our metric does not rely on input or output modality and the choice of backbone networks. Both modalities can be visual or textual for a given context since the cross-mutual information is based on the defined probability distributions. However, we may be not available to the pretrained model with the CLIP-scale (large data, high computing) for video or audio data, the applications might be limited for the other scenarios.

---

### Official Review · Reviewer_Ezue · 2022-07-11

**Rating:** 6
**Confidence:** 3
**Soundness:** 3 good
**Presentation:** 3 good
**Contribution:** 3 good

**Summary:**

This paper introduced Mutual Information Divergence (MID), a new metric for measuring the quality of image-to-text and text-to-image generation. Similar to model-based metrics like FID, MID relied on a pre-trained backbone (CLIP) to extract features which are used for computing scores. The score is computed by measuring the point-wise MI between to two feature spaces (with Gaussian assumption). Results show that MID can better measure the quality of image-text pairs by comparing against existing metics on augmented dataset.

**Questions:**

The feature from the transformer-based CLIP model (layer-normed) seems to be contradicting the assumption that both X and Y are Gaussians, am I missing something or MID simply works regardless of the normalization of the input feature.


**Limitations:**

Besides numerical limitation, there are other limitations not discussed in the paper. For example, as a trade-off for leveraging pre-trained backbone model, this metric is might not be applicable to image/text that mismatches the pre-training scenario of the backbone model. (e.g. different language, different shape of image) In other word, this metric is only feasible for high-resource problems.

**Strengths And Weaknesses:**

Pros
+ The metric is simple and straightforward (in a good way) by combining powerful multi-modal backbone that allows cross-modal relation to be considered and MI to be computed rather easily. (my only concern is the Gaussian assumption, see Questions)
+ The most important contribution is perhaps the fact that this metric can be applied regardless of the input/output modality and the choice of backbone can potentially be flexible as well.
+ Experiments showed that MID is better than existing metrics and aligns nicely to human evaluation.
+ The authors considered many different aspects (such as consistency, overfitting backbone, hallucination sensitivity) of designing a new metric and conducted experiment respectively to show that MID is robust.

Cons
- Only few models are considered for evaluation. It would be nice to see the results when applying MID to evaluate more models since metrics are designed to be used to compare models.
- While the advantage over simple metrics like BLEU is clear, the disadvantage is not discussed throughout in the paper. (see limitations)

---

> ### Author Response · Authors · 2022-08-02
> **Author feedback for Reviewer Ezue**
>
> **1. Gaussian Assumption.** After l2-normalization of CLIP features, these features become unit vectors lying on the hyperspherical surface. In our preliminary experiment, we empirically found that the feature samples projected on the 1st to 3rd eigenvectors of the covariance matrix are (near-)Gaussian for both visual and textual features. We conjecture that the features are following Gaussian in a sufficiently local area. One may think of a tangent space to calibrate the points far from the center; however, these outliers may not need such extra work for accurate evaluation in our scenario.
>
> **2. Domain Generalization.** We understand the “considering image/text mismatched pre-training scenario" as the domain generalization capability of the MID. Since we measure the multimodal alignments by relatively measuring the divergence from the pairs of real images and texts, we believe our metric would retain the majority of performances; but we do not provide empirical validations, which is the limitation of the current work. Specifically, different languages or drastic change of image domain would result underperformance, but it may be shared among the CLIP-based metrics, e.g., CLIP-S. Notably, since our method is not designed to particular pre-trained backbone models, we could trivially employ domain-specific backbones or state-of-the-art few-shot learning techniques for low-resource scenarios.
>
> **3. More Models for Evaluation.** We refer to Table 9 in Appendix where we include the MID reports comparing with the other metrics and various generative models, i.e., GLIDE, AttnGAN, DM-GAN, OP-GAN, DF-GAN, VQ-Diffusion, and LAFITE. In Sec 4.1, L225-226 points to this table, for your information.

---

### Official Review · Reviewer_W8Mn · 2022-07-13

**Rating:** 6
**Confidence:** 3
**Ethics Flag:** Yes
**Soundness:** 2 fair
**Presentation:** 2 fair
**Contribution:** 3 good

**Summary:**

This paper presents an automated metric based on mutual information divergence for multimodal generative models. The method exploits gaussian mutual information framework and cross-mutual information. It uses image and text encoders of CLIP to compute the mutual information divergence. The method is theoretically well motivated and presents promising empirical results.

**Questions:**

In section 4.1: there is an assumption being made where it says that fake images are generated by 'foiled' captions are inferior compared to fake images created using regular captions. How true is this assumption? FOILed captions are not necessarily "wrong" captions. They can indeed be plausible and hence the generated images can be of better quality. Some validation for this would be useful.

**Ethics Review Area:**

["Inappropriate Potential Applications & Impact  (e.g., human rights concerns)"]

**Limitations:**

The paper does not present critical limitations of the metric (such as reliance of CLIP/lack of validity over languages) or issues relating potential dual use of the metric.

**Strengths And Weaknesses:**

Strengths:
1. The metric is theoretically well motivated and is consistent across datasets and domains.
2. The proposal in the paper correlates well with human judgement across several datasets.
3. Thorough comparison to previous work.


Potential concerns:
The results currently do not support if this metric is absolute - it is not clear how reliant the empirical results are with CLIP. What if there was no CLIP?
Missing reference: VIFIDEL: Evaluating the Visual Fidelity of Image Descriptions by Madhyastha et al (ACL 2019)

---

> ### Author Response · Authors · 2022-08-02
> **Author feedback for Reviewer W8Mn**
>
> **1. How reliant are the empirical results on CLIP?** We believe our method is not strongly-coupled with the CLIP; however, we still need good multimodal features. Let us elaborate. In Sec 4.1.1, “Inspecting possible over-fitting with the CLIP features” partly explores this. The CLIP consists of visual and textual encoders and the InfoNCE loss maximizing the mutual information of multimodal features. Multiple versions of the pretrained CLIP vary the architectural type of visual backbone networks. We found that the normalized scores are generally consistent across generative models and the backbones, especially for our metric (see Fig. 3). Although we did not explore the textual encoders, we do not expect largely different results. However, it is worth noting that we encouraged the usage of two feature extractors having multimodal alignments for accurate metrics (See the last paragraph of Sec 3.1; L125-129). Thus, it is an interesting question whether a non-CLIP-based feature works for MID. For example, we may use unimodal encoders, e.g., the ViT pretrained on ImageNet (Dosovitskiy et al., 2020) and the sentence-BERT (Reimers & Gurevych, 2019).
>
> **2. A Related Work.** Thanks for the reference. We will add VIFIDEL (Madhyastha et al., 2019) for the comparison in the image captioning task.
>
> **3. Inferior Foiled.** Our foiling technique is based on Shekhar et al. (2017; see https://foilunitn.github.io). They exploit 73 out of 91 MS-COCO categories, excluding multi-word expressions (e.g., traffic light). Since the MS-COCO categories are distinctive, we believe it *severely* hurts the content of captions. For this reason, we do not think the generated images using the foiled captions can be of any better quality than those using the original captions when we match the generated images with the original captions. In this way, we intend that the foiled fake images are deliberately mismatched with the corresponding original captions. Please see Sec 4.1, L164. *“the generated images”* includes the foiled fake along with the fake images.)
>
> **4. Ethical considerations.** We acknowledge the raised issue that the pretrained CLIP potentially induces the risk of fairness, accountability, and transparency. We plan to include a dedicated paragraph in the main paper. For the detail, please refer to the comment under the *Reviewer wkz2*.

---

### Review · Ethics_Reviewer_Vvz8 · 2022-07-27

**Recommendation:** Review this paper as-is

**Ethics Review:**

Only one reviewer flagged an ethical issue, and they didn't elaborate on their concern. This is a technical paper on a well-established task (image captioning), and while I guess it would be possible to contrive of a setting where such captions produced harm, the same is true for any task. I don't see any ethical issues with this paper.

---

### Review · Ethics_Reviewer_oJLc · 2022-08-03

**Recommendation:** Again, there are no ethical issues here.

**Ethics Review:**

The paper proposes a new metric for evaluating text to image generation and image captioning models. The effectiveness of the proposed metric is then demonstrated through extensive experiments on various datasets. The paper in its nature does not raise any ethical concerns and this is indicated by the majority of the reviewers.

---

### Author Response · Authors · 2022-08-08
**A gentle encouragement to engage in discussion**

Politely, we remind you that the discussion phase is ending soon. We want to know if the raised concerns may be resolved after the following author's feedback. We have carefully read the reviewers' thoughtful comments and encouraging suggestions, and provided constructive reflections.

---

### Meta-Review · Area_Chair_pmUS · 2022-08-24

**Recommendation:** Accept
**Confidence:** Certain

**Metareview:**

The paper studies the evaluation metric for multimodal generation models. The authors propose a method MID based on estimating mutual information of visual and text embeddings at sample and distribution level. From experiments, the MID correlates with human evaluation on multiple tasks (text-to-image and image captioning). The authors provide theoretical intuition and analysis of MID and relation to other divergence scores. Experiments are solid and convincing. The reliance on CLIP is discussed though other multimodal encoders than CLIP are not evaluated in experiments. Author discussion with reviewers are helpful to better understand the paper.

Overall, it is a solid paper with a clearly described, simple, and effective method.

**Award:**

No

---

### Decision · Program_Chairs · 2022-09-14

Accept